# Huntington's disease phenotypes are improved via mTORC1 modulation by small molecule therapy

**Sophie St-Cyr**[1], **Daniel D. Child**[1,2], **Emilie Giaime**[3], **Alicia R. Smith**[1], **Christine J. Pascua**[4], **Seung Hahm**[3], **Eddine Saiah**[3]*, **Beverly L. Davidson**[1,2]*

**1** The Raymond G. Perelman Center for Cellular and Molecular Therapeutics, The Children's Hospital of Philadelphia, Philadelphia, PA, United States of America, **2** The Perelman School of Medicine, The University of Pennsylvania, Philadelphia, PA, United States of America, **3** Navitor Pharmaceuticals Inc., Cambridge, MA, United States of America, **4** Division of Cardiology, The Children's Hospital of Philadelphia, Philadelphia, PA, United States of America

* davidsonbl@chop.edu (BLD); esaiah@navitorpharma.com (ES)

## Abstract

Huntington's Disease (HD) is a dominantly inherited neurodegenerative disease for which the major causes of mortality are neurodegeneration-associated aspiration pneumonia followed by cardiac failure. mTORC1 pathway perturbations are present in HD models and human tissues. Amelioration of mTORC1 deficits by genetic modulation improves disease phenotypes in HD models, is not a viable therapeutic strategy. Here, we assessed a novel small molecule mTORC1 pathway activator, NV-5297, for its improvement of the disease phenotypes in the N171-82Q HD mouse model. Oral dosing of NV-5297 over 6 weeks activated mTORC1, increased striatal volume, improved motor learning and heart contractility. Further, the heart contractility, heart fibrosis, and survival were improved in response to the cardiac stressor isoprenaline when compared to vehicle-treated mice. Cummulatively, these data support mTORC1 activation as a therapeutic target in HD and consolidates NV-5297 as a promising drug candidate for treating central and peripheral HD phenotypes and, more generally, mTORC1-deficit related diseases.

## Introduction

Huntington's Disease [HD] is a fatal, autosomal dominant, neurodegenerative disease caused by a CAG repeat expansion in exon 1 of the *Huntingtin* gene [1]. This mutation translates as a poly-glutamine expansion in the ubiquitously-expressed mutant Huntingtin [mHTT] protein and promotes molecular and cellular dysfunction [2]. Symptoms include cognitive decline, progressive involuntary movements (chorea), behavioral abnormalities, muscle wasting, and death within 10 to 15 years of symptom onset [3, 4]. The prevalence of HD in Europe and the United States is 5–7 per 100,000, making it among the most common dominantly inherited neurodegenerative diseases [4]. Medium spiny neurons, constituting 90–95% of striatal neurons, are especially vulnerable to HD-related degeneration. There is currently no effective treatment to halt or delay disease progression, though considerable research effort has been

**Data Availability Statement:** All relevant data are within the manuscript and its Supporting Information files.

**Funding:** BLD received sponsored research support from Navitor pharmaceuticals. EG, SH, and ES are employees of Navitor Pharmaceuticals. The funders shared a role in study design, data collection, decision to publish and preparation of the manuscript.

**Competing interests:** BLD received sponsored research support from Navitor pharmaceuticals. EG, SH, and ES are employees of Navitor Pharmaceuticals. This does not alter our adherence to PLOS ONE policies on sharing data and materials.

devoted to ending this devastating disease. Recently, clinical trials using gene silencing technologies with antisense oligonucleotides delivered to CNS failed to show target engagement or worsened disease readouts [5, 6]. As such, investigation of additional therapeteutics is warranted. Moreover, as HTT is expressed ubiquitously and affects peripheral tissues [7, 8], a comprehensive therapeutic approach that targets all tissues would be ideal.

Heart disease is the second cause of mortality in HD [9, 10], and heart failure incidence is increased in HD patients compared to unaffected age-matched controls [11, 12]. HD patients also present a smaller heart size [13]. Premanifest HD patients also show higher cardiovascular risk factors compared to age- and gender-matched controls [14]. These findings are recapitulated in multiple HD mouse models expressing mutant HTT fragments ubiquitously, including smaller heart size, impaired cardiac function through dilated ventricles, fibrosis, and increased apoptosis of cardiomyocytes [15–19]. These phenotypes are exacerbated by chronic β-adrenergic agonist stimulation and by age, suggesting diminished physiologic cardiac reserve [19–21].

The mTOR complex 1 [mTORC1] is a serine-threonine kinase that integrates multiple environmental cues such as nutrient and oxygen availability, and, when activated, promotes cell growth and metabolism through enhanced protein synthesis and multiple other downstream effectors. Conversely, inactivation of the mTORC1 pathway is associated with upregulated macroautophagy and cell survival mechanisms. Amino acid availability directly affects protein synthesis rate and provides essential upstream signals for mTORC1 pathway activation. Specifically, Sestrin1 and 2 are upstream mTORC1 pathway effectors that detect and bind the proteogenic amino acid leucine. Leucine binding induces Sestrin dissociation from the multiprotein complex GATOR2, inducing a signalling cascade that activates mTORC1 [22, 23]. mTORC1 enhances protein synthesis and mRNA translation initiation through the phosphorylation of the positive effector ribosomal protein S6 [S6] and the negative regulator eukaryotic translation initiation factor 4E-binding protein 1 [4E-BP1], both constituting accurate readouts of mTORC1 activation [24, 25].

In the brain, mTORC1 is neuroprotective and promotes myelination, axon growth and regeneration [26–28]. In the absence of the mTORC1-activating subunit <u>Ras h</u>omolog <u>e</u>nriched in the <u>b</u>rain [Rheb] or <u>s</u>triatum [Rhes], there is decreased cortical thickness and deficits in myelination [29]. Further, Rhes KO mice have motor coordination deficits associated with striatal health [30], similar to HD. In the heart, the mTORC1 pathway is a master regulator of growth and stress response. Cardiac-specific Rheb KO mice have reduced mRNA translation and cardiomyocyte growth, impaired cardiac function during the early postnatal period, and shortened life span. Ablation of the inhibitor 4EBP1 in Rheb KO animals restores cardiomyocyte size, suggesting that mTORC1's effect on heart size is through regulation of translation initiation [31, 32]. In adult hearts, mTORC1 activity mediates the heart's response to chronic stress. Cardiac-specific Raptor deletion, an essential mTORC1 subunit, causes maladaptation in response to pressure overload, resulting in dilated cardiomyopathy and heart failure without the initial adaptive responses [33]. Conversely, overexpression of the mTORC1 catalytic subunit mTOR attenuates pressure overload-induced fibrosis and reduces the associated inflammatory response [34]. Furthermore, the mTORC1 inhibitor rapamycin prevents appropriate cardiac compensation to volume and pressure overload and increases mortality in heart pathology models [35, 36]. Interestingly, the cardiac dysfunctions observed in mTORC1 pathway dysregulation are consistent with those noted in HD animal models.

The mTORC1 pathway is dysregulated in HD patient striatum on post-mortem analysis, a phenotype recapitulated in multiple mouse models. Additional mTORC1 pathway suppression with small molecule inhibitors accelerated disease phenotypes in multiple HD mouse models [37]. Conversely, genetic restoration of mTORC1 activity in the mHTT fragment mouse model N171-82Q alleviated neuropathology, striatal atrophy, and motor deficits [38]. In the

heart, mTORC1 pathway activation was similarly reduced in both N171-82Q mice and in the knock-in mouse model zQ175, with constitutively active Rheb restoring stress adaptation and reducing mortality [19]. These data suggest that mTORC1 pathway activation can improve HD-associated deficits. However, mTORC1 activity requires a delicate balance, as chronic activation is associated with multiple adverse conditions, including but not limited to tuberous sclerosis, cortical dysplasias, and certain cancers [24]. Consequently, genetic overactivation is useful as a proof-of-concept demonstration, but a valid therapeutic approach requires the ability to fine-tune the degree of mTORC1 activity modulation.

We previously reported a small molecule, NV-5138, that could induce mTORC1 in brain and peripheral tissues. NV-5138 binds Sestrin2 and transiently activates mTORC1 in peripheral tissues and CNS [39, 40]. Here, we tested oral dosing of NV-5297, a closely related analog of NV-5138, on striatal and heart pathologies as well as motor behavior in N171-82Q HD mice. Our data show that NV-5297 improves N171-82Q HD heart function under physiological and stress conditions, ameliorates motor deficits, and extends animal survival under cardiac stress conditions. Cumulatively, our data support a role for transient mTORC1 activation as a treatment for both the CNS and peripheral symptoms of HD.

## Materials and methods

### Animals

This study was carried out in strict accordance with the recommendations in the Guide for the Care and Use of Laboratory Animals of the National Institutes of Health. N171-82Q animals' protocols were approved by the Animal Care and Use Committee at the Children's Hospital of Philadelphia and IACUC (licence #1358). Experiments performed at Biomodels LLC were approved by Biomodels LLC'IACUC (16–061402) and the Office of Laboratory Animal Welfare. Experiment performed at Chempartner were approved by AAALAC (license #001321). The N171-82Q HD murine model strain express a human N-terminal truncated HTT with 82 polyglutamine repeat driven by a mouse prion protein promoter [41]. Breeder hemizygous males were obtained from Jackson Laboratories and animals were bred in-house and maintained on a B6C3F1/J background. Mice were genotyped using primers specific for human HTT: F 5'-ATG GCG ACC CTG GAA AAG CTG-3' and R 5'-TCG GTG CAG CGG CTC CTC-3'. No more than two hemizygous or wild-type males per litter per experimental group were used. N171-82Q mice and C57B/6 mice were housed on a 12-hr light/dark cycle with light on at 6:15AM and *ad libitum* access to food and water in an enriched and temperature-controlled environment. Animals were randomly assigned to treatment groups for all experiments at the Children's Hospital of Philadelphia, Biomodels LLC or at Chempartner.

### NV-5297 preparation

The complete protocol for synthesis of NV-5297 [(S)-2-amino-5,5,5-trifluoro-4,4-dimethyl-pentanoic acid] is available in the S1 File. NV-5297 was dissolved in the vehicle (0.5% Methylcellulose and 0.1% Tween80) and was administered by oral dosing at 160 mg/kg body mass around 7AM daily. The control group was dosed orally with the vehicle daily in the same fashion as the NV-5297 treated group.

### Pharmacokinetic analysis of NV-5297 in mice

To determine oral bioavailability, male C57BL/6 were dosed i.v. at 1 mg/kg and orally at 5 mg/kg with NV-5297 in the vehicle (n = 3 per time point per group). Mice were given free access to food and water for both i.v. and orally dosed groups. After dosing, tail-vein blood was

collected at the indicated time points into K2EDTA tubes and centrifuged at 2,000$g$ for 5 minutes to collect serum. Levels of NV-5297 were quantified via LC-MS/MS. WinNonlin V 6.2 statistics software (Pharsight Corporation, California, USA) was used to generate pharmacokinetics parameters using non-compartmental model.

## Study design

Animals were assigned to an experimental group so that their pre-treatment behavioral performance on the accelerated rotarod, forelimb grip strength and baseline echocardiography measurements were similar. All animals tested were hemizygous N171-82Q males and some tissue were collected from wild-type males from the same strain. Female mice were excluded from the studies due to wide variations in mTORC1, likely activity related to estrous cycling. The experiments were performed with at least two different cohorts of animals, and all cohorts showed the same trend.

The initial trial consisted in one week of oral dosing in 12 weeks old N171-82Q males. The chronic 6 weeks treatment (Veh n = 15, NV-5297 n = 14) took place from 6 to 12 weeks of age. The cardiac stress took place in animals from 10 to 12 weeks of age (Veh-Sal n = 9, NV-Sal n = 10, Veh-Iso n = 16, NV-Iso n = 11).

Animals were sacrificed by live decapitation one hour after the last oral dosing. Whole hearts were excised, blot-dried and weighed on an analytical balance. The heart was bisected transversely and the apical halves was preserved for cardiomyocyte and fibrosis staining. Left tibias were isolated and measured with an electronic caliper. The brain left hemisphere was preserved for MSN size measurement. The brain right hemisphere was microdissected and the medial pre-frontal cortex, striatum, cortex, hippocampus and cerebellum were collected.

## Antibody list for western blotting

Primary antibodies used for Western Blot were polyclonal rabbit anti-phosphorylated p70 S6 Kinase (Thr389; Cell Signaling Technology; #9205), monoclonal rabbit anti-p70 S6 Kinase (Cell Signaling Technology; #2708), monoclonal rabbit anti-phosphorylated S6 ribosomal protein (Ser240/244; Cell Signaling Technology, #5364), monoclonal rabbit anti-S6 ribosomal protein (Cell Signaling Technology, #2217), monoclonal mouse anti-S6 ribosomal protein (Cell Signaling Technology, #2317), monoclonal rabbit anti-phosphorylated Akt (S437; Cell Signaling Technology, #4060), monoclonal rabbit anti-Akt (Cell Signaling Technology, #4691), monoclonal rabbit anti-phosphorylated 4E-BP1 (Thr37/46; Cell Signaling Technology, #2855), monoclonal rabbit anti-4E-BP1 (Cell Signaling Technology, #9644), monoclonal rabbit anti-phosphorylated mTOR (Ser2448; Cell Signaling Technology, #5536), polyclonal rabbit anti-mTOR (Cell Signaling Technology, #2972), polyclonal rabbit anti-DARPP32 (Thermo Fisher Scientific, #720203), polyclonal rabbit anti-Sestrin 1 (Sigma, #SAB4501207), monoclonal rabbit anti-Sestrin 2 (Cell Signaling Technology, #8487), monoclonal mouse anti-Sestrin 3 (Sigma, #WH0143686M2), polyclonal rabbit anti-Nprl3 (Atlas, #HPA011741), polyclonal rabbit anti-actin (Sigma-Aldrich, #A2066), monoclonal anti-Flag M2 affinity gel (Sigma-Aldrich, #A2220), and monoclonal mouse anti-tubulin (Sigma-Aldrich, #T5168) and were used at a 1:1000 dilution. Secondary antibodies used for WB were goat anti-rabbit IRDye 800CW (#926–32211), donkey anti-mouse IRDye 680RD (#926–68072), and goat anti-mouse IRdye680RD (#925–68070) from LI-COR. Antibodies were previously validated [40].

## Cell culture

All cell lines were cultured in DMEM (Corning, #MT10013CV) supplemented with 10% heat-inactivated FBS (Gibco, #16140–071). All cell lines were maintained at 37˚C and 5% $CO_2$. For

amino acid starvation, cells were rinsed once with and incubated Amino acid-free RPMI (US Biologicals, #R8999-04A) supplemented with 10% dialyzed FBS (Gibco, #26400044) for 50 min followed by treatment with NV-5297, leucine, or vehicle for 10 minutes. For leucine starvation, cells were rinsed once with and incubated in leucine-free DMEM (AthenaES, #0420) supplemented with 10% dialyzed FBS (Gibco, #26400044) for 50 minuted followed by treatment with NV-5297, leucine, or vehicle for 10 minutes.

## Intact cells-based protein-protein interaction assay

Flag-WDR24 293T cells were washed and placed in leucine-free DMEM supplemented 10% dFBS for 50 minutes followed by treatment with NV-5297, leucine or vehicle for 10 minutes. Cells were rinsed once with ice-cold PBS and immediately lysed with Triton lysis buffer (1% Triton, 10 mM β-glycerol phosphate, 10 mM pyrophosphate, 40 mM Hepes pH 7.4, 2.5 mM MgCl2 and 1 tablet of EDTA-free protease inhibitor). The cell lysates were cleared by centrifugation at 13,200 rpm at 4˚C in a microcentrifuge for 10 minutes. Prior to immunoprecipitations, an aliquot of lysate was taken for SDS-PAGE and immunoblotting for pS6K1, S6K1 and tubulin. For anti-FLAG immunoprecipitations, the FLAG-M2 affinity gel was washed 3 times with Triton lysis buffer, and 30 μl of a 50/50 slurry of the FLAG-M2 affinity gel was then added to approximately 2 mg of clarified cell lysates and incubated with rotation for 3 hours at 4˚C. Following immunoprecipitation, the beads were washed one time with Triton wash buffer containing 500 mM NaCl. Immunoprecipitated proteins were denatured, resolved by SDS-PAGE, and analyzed by immunoblotting.

## Lysate-based protein-protein interaction assay

Flag-WDR24 293T cells were washed and placed in amino acid-free RPMI for 50 minutes. Cells were then rinsed once with ice-cold PBS and immediately lysed with Triton lysis buffer. The cell lysates were cleared by centrifugation at 13,200 rpm at 4˚C in a microcentrifuge for 10 minutes. The anti-FLAG immunoprecipitations were performed as described above. Immunoprecipitated proteins (IPs) were resuspended in cytosolic buffer (40 mM HEPES pH 7.4, 140 mM KCl, 10 mM NaCl, 2.5 mM MgCl2, 0.1% TritonX-100) containing vehicle, NV-5297 or leucine at the indicated doses for 10 minutes. After compound incubation, IPs were collected by centrifugation, denatured, resolved by SDS-PAGE, and analyzed by immunoblotting.

## Behavioral testing

The accelerated rotarod (model 47600; Ugo Basile, Comerio, Italy) test was carried out as previously described [42]. Mice were tested at 5 (baseline) and 12 weeks of age. Briefly, mice were habituated to the test room for an hour. On the first day, mice were trained on the rotarod for 5 minutes at 4 rpm. Mice were then tested in three trials per day (with at least 30 minutes of rest between trials) for four consecutive days. For each trial, the rotarod accelerated from 4 to 40 rpm over 4 minutes with constant speed at 40 rpm for an additional minute. Trials were stopped at 300 seconds. Latency to fall (or two consecutive rotations without running) was recorded for every trial. For grip strength, mice were tested at 5 (baseline), 9 and 12 weeks of age. Mice were habituated to the test room for 1 hour. The mouse held a pull bar with both paws and are then pulled horizontally until they let go of the pull bar. Forelimb grip strength at the peak tension was measured five times per mice (Columbus Instrument). There were at least 30 minutes of rest between trials. The grip strength is calculated as the average of the four highest grip strength recorded.

## Echocardiography

Mice were anesthetized with 2% isoflurane and restrained on a stage with electrocardiogram (EKG) sensors. Transthoracic echocardiograms were performed using the Vevo 3100 Imaging System with a linear array MX550D transducer. Images were acquired by an experienced echo-cardiographer and then analyzed offline using the Vevo LAB analysis software. Parasternal long axis views and short axis views were used to assess left ventricular function. Measurements were performed to evaluate left ventricular dimensions during both systole and diastole of the cardiac cycle. The Vevo LAB left ventricle [LV] analysis tool, LV trace, was used to calculate cardiac output [CO] (heart rate · stroke volume), fractional shortening [FS] (ventricle diameter at the end of the diastole–ventricle diameter at the end of the systole), ejection fraction [EF] (percentage of blood ejected by the left ventricle by heart beat), stroke volume (SV) (blood volume ejected from the left ventricle per beat), heart rate [HR] (bpm), left ventricular mass [LVmass], dimensions and volumes. Both imaging and analysis were performed by an operator blinded to the mice experimental treatment.

## Protein quantification and analysis

Specific regions of the brain and heart were homogenized using a Next Advance Bullet Blender Storm in Lysis Buffer (Cell lysis buffer: 1% Triton X-100, 50 mM HEPES pH 7.4, 100 mM sodium chloride, 2 mM EDTA, 10 mM Beta-glycerophosphate, 10 mM Sodium pyrophosphate, and 1 protease inhibitor) at 4˚C. Lysates were cleared by centrifugation at 13,200 rpm at 4˚C for 10 minutes. Equal amounts of total protein from each sample were denatured and, resolved by SDS-PAGE. Membranes were imaged using the LI-COR imaging system. Proteins levels are normalized to tubulin, actin or GAPDH levels and further normalized to vehicle treated mice for each tissue.

## Histology analyses

Mice were perfused with PBS, and tissues were fixed in 4% paraformaldehyde for at least 48 hours. Brains were cryoprotected in a 30% sucrose solution for 24 hours and cryosectioned to 40 μm on a Leica SM2010R microtome. Each section through the striatum was collected and floated out in 30% ethylene glycol/15% sucrose solution sequentially in a 12-well plate. Once sections had been placed in each well, deeper sequential sections were added to wells in the same order, leading to each well containing 7 sections, 480 μm apart. Hearts were bisected transversely, and the apical halves were processed and embedded in paraffin. Heart tissue was sectioned to 5 μm on a Microm HM 355S (Thermo) microtome. For stereology, all sections from one well of the 12-well plate were stained as floating sections for DARPP-32 as previous described [38]. Briefly, sections were incubated in a 0.1 M sodium meta periodate solution to block endogenous peroxidase, then blocked in 5% goat serum and incubated with DARPP-32 primary antibody (1:400; Cell Signaling, #2306) overnight. Sections were then incubated in biotinylated goat anti-rabbit secondary antibody (1:200; Vector Laboratories), followed by incubation with peroxidase-labelled VectaStain Elite ABC kit (Vector Laboratories). Signal was developed with DAB. Sections were mounted on slides and imaged using a Leica DM6100 light microscope. All analyses were performed in ImageJ.

Striatal volume were estimated by the Cavalieri method [43] on low-power digital images of DARPP-32-stained sections in ImageJ. A 5000 μm$^2$ area-per-point grid was superimposed randomly over the image, and number of points overlapping the striatum in each section (5 per animal) was counted. The volume was then calculated using the formula $V = \sum p \cdot t \cdot A(p)$, where $V$ = estimated volume, $p$ = points per section, $t$ = space between sections, and $A(p)$ = area-per-point. The volumes were aggregated based on treatment group.

Average MSN area was calculated from high-power digital images of DARPP-32-stained sections in ImageJ. Four sections were randomly selected from among the seven per brain, a grid was superimposed over the images to select a random subset of MSNs, and the area of at least 80 MSN cell bodies was measured manually within each section. At least 700 cells were measured per brain, and the average MSN area was determined per treatment group.

Cardiomyocyte area was measured in wheat germ agglutinin-labelled heart sections as previously described [19]. Briefly, sections were cleared and rehydrated through xylene and ethanol, then labelled with WGA conjugated to Alexa Fluor 594 (8 μg/ml, Biotium) per manufacturer's protocol. Sections were counterstained with Hoechst stain (2 μg/ml, Thermo Scientific). High-power digital images were collected such that the entirety of the left ventricular free wall was represented. A grid was superimposed over each image in ImageJ to randomly select cardiomyocytes sectioned perpendicular to the long axis, and the area of those cells was measured manually. At least 240 cardiomyocytes were measured per section. Cardiomyocyte areas were averaged for each treatment group.

The percent of area occupied by fibrotic tissue was calculated as previously described [19]. Briefly, heart sections were cleared and rehydrated through xylene and ethanol. Sections were counterstained with Weigert's haematoxylin (Sigma) per manufacturer's protocol. Collagen was labelled by staining to equilibrium with Picrosirius red (0.1% Direct Red 80, Sigma, in saturated picric acid, Sigma). Bright field and corresponding polarized light digital images were taken such that the entirety of the left ventricular free wall was represented. Image threshold was adjusted to highlight either the entire tissue (bright field) or the fibrotic tissue (polarized) in ImageJ, and percent area occupied by fibrotic tissue was calculated from these images. Large vessels were excluded from the analysis.

## Cardiac stress induction

Isoprenaline (Sigma) was dissolved in sterile PBS and administered by continuous infusion using implantable mini-osmotic pumps (Alzet) at a dose of 30 mg/kg body mass/day. Pumps were filled and primed in a sterile environment and inserted under 2.5% isoflurane anesthesia into the right dorsal subcutaneous space caudal to the scapulae.

## Statistics

All statistics were conducted using the Prism 8 software. Group's normality was evaluated with the Kolmogorov-Smirnov test. All the significant effects reported in this manuscript have large effect sizes with one comparison having a medium effect size (Fig 2B, Day 1 vs 2 comparison).

Comparing the tissue- and brain-region specific pS6 protein levels in C57BL6 mice was performed using a two-way repeated measure ANOVA with Sidak's multiple comparison tests, with NV-5297 treatment and specific tissue as the main factors.

Western blot protein quantifications were analyzed by Welch's ANOVA test with Dunnett's T3 multiple comparisons test for normal distributions with groups of unequal variances (one-week exposure heart pS6), Kruskal-Wallis test with Dunn's multiple comparaison tests for non-normal distributions (One-week exposure striatum pS6), one-tailed unpaired Student's t-test for normal distributions (Chronic exposure: Striatum: pmTOR/mTOR and DARPP-32, Heart: pmTOR and p4E-BP1), Student's t-test with a Welch's correction in the case of normal distributions with groups of unequal variances (Chronic exposure: Striatum: pS6, Heart: pmTOR/mTOR) or a Mann-Whitney test for non-normal distributions (Chronic exposure: Striatum: pAkt/Akt and p4E-BP1, Heart: pS6 and pAkt/Akt) with the NV-5297 treatment as the main factor. A Spearman correlation was performed between pS6/S6 and DARPP-32 levels in the striatum. Under cardiac stress, the treated groups were compared by Mixed-effects

models with Sidak's multiple comparisons tests with NV-5297 treatment and cardiac stress as the main factors.

The accelerating rotarod latencies to fall were analyzed by a two-way repeated-measure ANOVA with Sidak's multiple comparisons test with NV-5297 treatment and time as the main factors in the chronic treatment experiment. In the cardiac stress experiment, the baseline rotarod performance was compared by a Mixed-effects model was used with NV-5297 treatment and cardiac stress and time as the main factors.

Forelimb grip strength in the chronic exposure experiment were compared using a two-way repeated measure ANOVA with NV-5297 treatment and time as main effects followed by Tukey's multiple comparisons tests (specific for NV-5297 treatment effects) or Dunnett's multiple comparison tests (specific for time effects). In the cardiac stress experiment, at baseline, a Mixed-effects model was used with NV-5297 treatment and cardiac stress as the main factors.

During the chronic NV-5297 treatment experiment, the mice weight was compared by a two-way repeated measure ANOVA with Sidak's multiple comparisons test with NV-5297 treatment and time as the main effects. In the cardiac stress experiment, animal weight difference to baseline were compared by a three-way ANOVA with the NV-5297 treatment, cardiac stress and time as main effects. Following the finding of the significant interaction of NV-5297 treatment by cardiac stress and time effect in the three-way ANOVA, a Mixed-effect model with Sidak's multiple comparisons test was performed between the vehicle-saline- and vehicle-isoprenaline-treated mice with cardiac stress and time as the main effects.

The corrected heart weights after chronic NV-5297 treatment were compared using unpaired Kruskal-Wallis test with Dunn's multiple comparison tests with NV-5297 treatment as the main factor. The corrected heart weight under cardiac stress was compared using Mixed-effect model with Sidak's multiple comparisons tests with NV-5297 treatment and cardiac stress as the main factors.

The variation in heart function (CO, EF, FS, SV, HR) from baseline through to 6 weeks of daily NV-5297 treatment were compared by unpaired one-tailed Student's t-tests with NV-5297 treatment as the main factor. The heart function measurements (CO, EF, FS, SV, HR, LVmass) variation from baseline after one or two weeks of cardiac stress were compared using Mixed-effect models with Sidak's multiple comparisons tests with NV-5297 treatment and cardiac stress as the main factors.

The longevity curves under NV-5297 treatment and cardiac stress were compared using a Mantel-Cox test.

The striatal volume and medium spiny neuron size were compared using unpaired one-tailed Student's t-tests with NV-5297 treatment as the main factor.

The cardiomyocyte size and quantification of fibrosis indicates a significant difference by Mixed-effect model with Sidak's multiple comparisons tests with NV-5297 treatment and cardiac stress as the main factors.

## Results

### The synthetic leucine analog NV-5297 modulates the Sestrin2-Gator2 interaction and activates the mTORC1 pathway in vitro and in vivo

Sestrin2 binds to GATOR2 under leucine-depleted conditions, and the addition of leucine can disrupt this interaction and activate mTORC1 within minutes [44]. To characterize the activity of NV-5297 (Fig 1A), we first tested its ability to rescue the effects of leucine depravation in vitro, specifically its ability to bind Sestrin2, inhibit the GATOR2 complex and activate the mTORC1 pathway [39, 45]. In HEK293T cells starved of leucine for 50 minutes, NV-5297

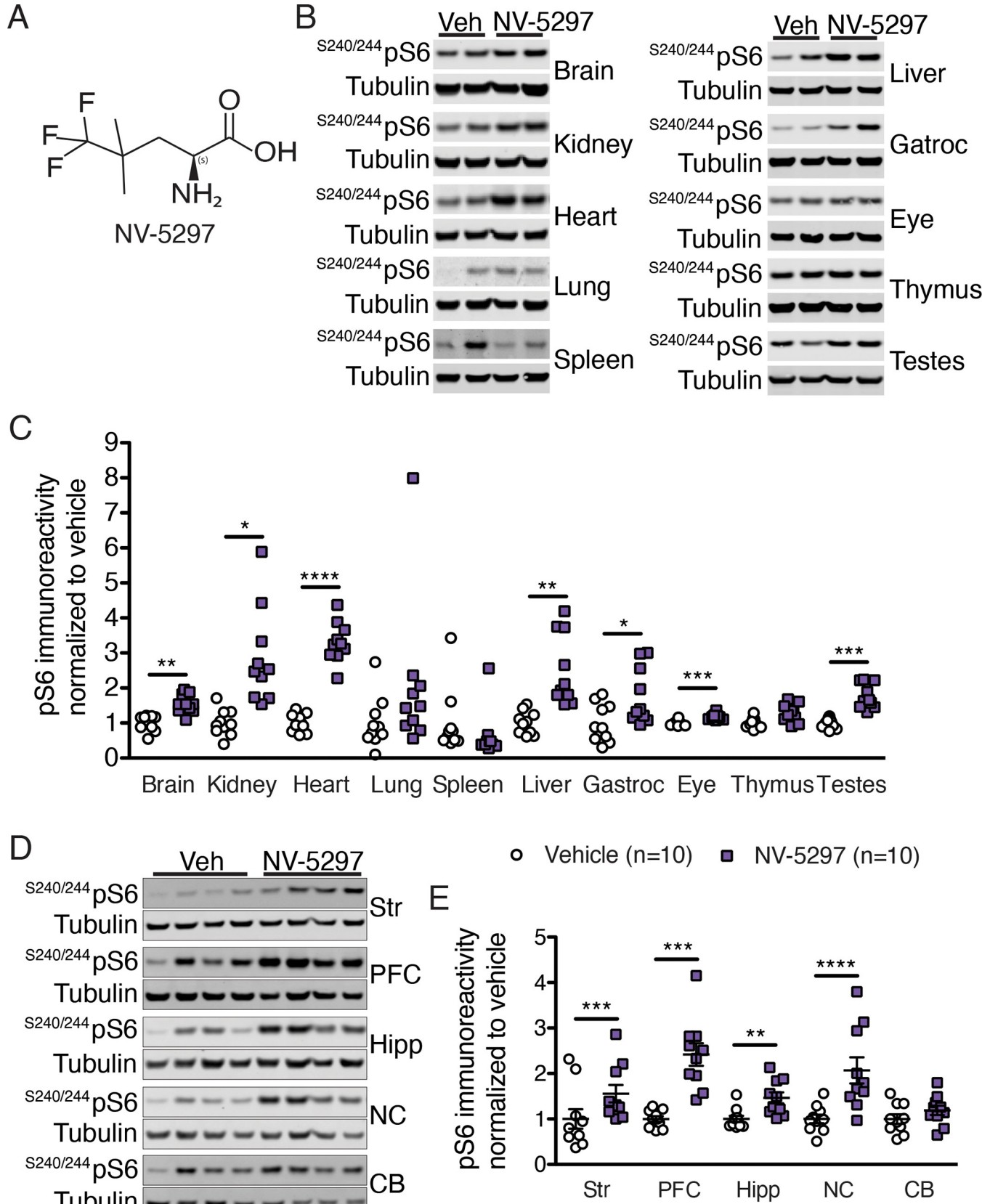

**Fig 1. NV-5297 is a leucine analog activating the mTORC1 pathway in the brain and peripheral tissues.** (A) NV-5297 chemical structure. Representative immunoblots of phosphorylated S6 ($^{S240/244}$pS6) from homogenized tissues from C57BL/6 mice orally dosed once with NV-5297 (160 mg/kg) and sacrificed 1 hour later (B) and quantification of the immunoblots normalized to loading controls and vehicle treated mice (n = 10 per treatment) (C). Representative immunoblots for the normalized $^{S240/244}$pS6 from various brain regions from homogenized tissues from C57BL/6 mice dosed orally once with NV-5297 (160 mg/kg) and sacrificed 1 hour later (D) and quantification of the immunoblots normalized to loading control levels and further normalized to vehicle treated mice (n = 10 brain per treatment) (E). Data are presented as mean ± SEM. CB: cerebellum, Gastroc: gastrocnemius muscle, Hipp: hippocampus, NC: nucleus accumbens, PFC: prefrontal cortex, Str: striatum, Veh: vehicle. *$P \leq 0.05$, **$P \leq 0.01$, ***$P \leq 0.001$ and ****$P \leq 0.0001$ (2-way repeated measure ANOVA with Sidak's multiple comparison tests).

addition activates mTORC1, as measured by increased phosphorylation of mTORC1's immediate downstream target 70S6K1 [pS6K1], without increasing total S6K1 levels [24] (S1A Fig).

Next, Sestrin2/GATOR2 protein-protein interactions in response to NV-5297 were measured using cell-free assays and *in vitro*. For this, HEK293T cells stably expressing flag-

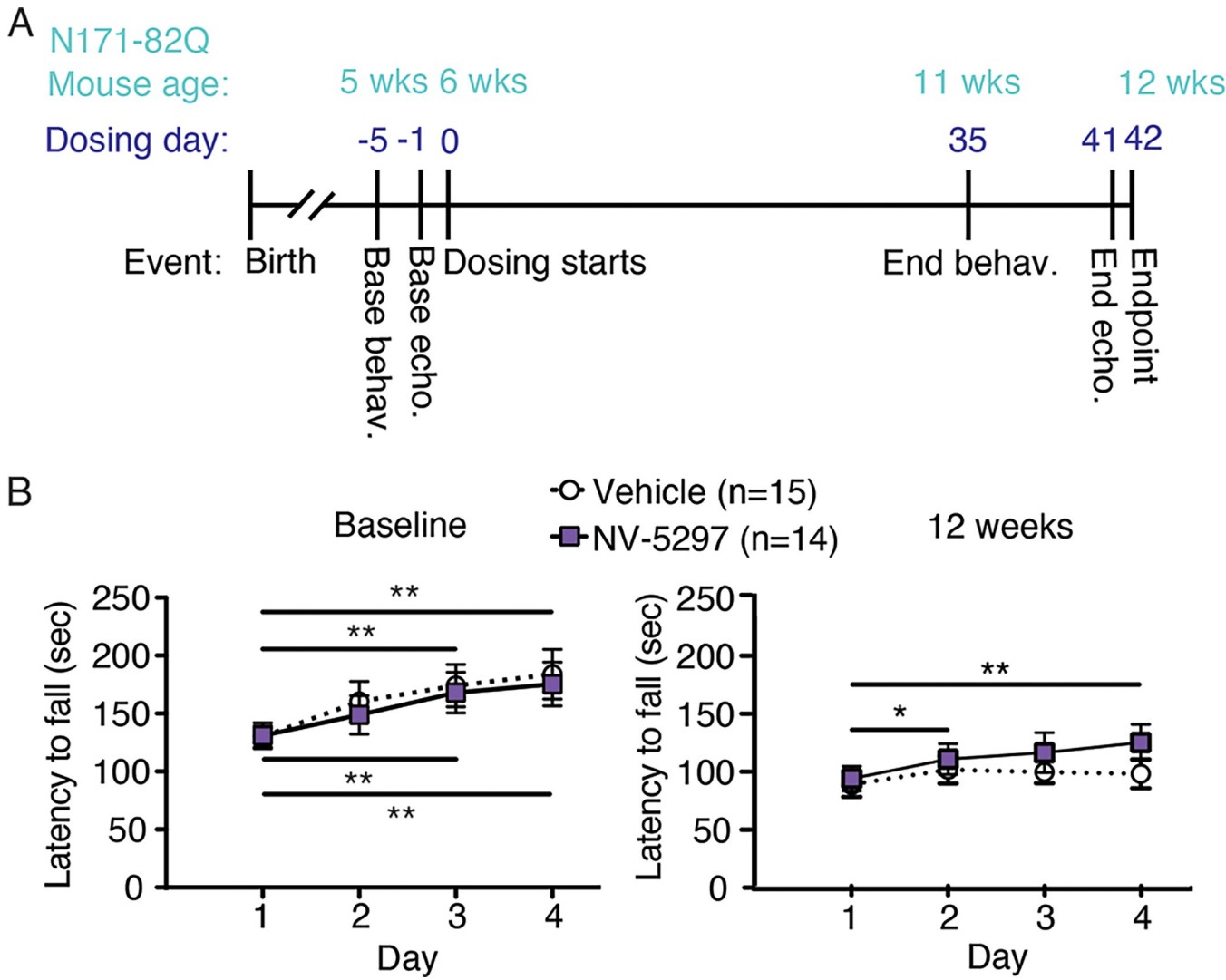

**Fig 2. Daily oral dosing of NV-5297 over 6 weeks improves the motor performance in N171-82Q males.** A Experimental timeline. 160 mg/kg NV-5297 is giving through oral dosing. B Latency to fall from the accelerating rotarod at baseline (6 weeks of age) and after 6 weeks of treatment (12 weeks of age) with either NV-5297 or vehicle. Statistical analysis identifies changes in performance within treatment groups over the course of the 4-day trial. Data are presented as mean ± SEM. *$P \leq 0.05$, **$P \leq 0.01$ (2-way repeated measure ANOVA with Sidaks's multiple comparison test).

WDR24, a component of GATOR2 [44], were used. The cell-free assays utilized Sestrin2/GATOR2 complexes immunoprecipitated from amino acid-starved cells with an anti-flag antibody. Addition of NV-5297 to either the immunoprecipitated Sestrin2/GATOR2 complexes or to cell cultures disrupted the Sestrin2/GATOR2 interaction in a dose-dependent manner, with levels of GATOR2-associated Sestrin2 inversely proportional to the NV-5297 dose (S1B Fig). NV-5297 treatment in leucine-starved cells also increased p70S6K1 in a dose-dependent manner (S1C Fig).

To ascertain if Sestrins and the GATOR pathway are required for NV-5297 activity, we tested mTORC1 activation in cells null for either all three Sestrin isoforms [SesnTKO] or for the GATOR1 component Npr13 [Npr13KO] [44]. Treatment with NV-5297 or leucine activates mTORC1 in leucine-starved HEK293T cells in a dose-dependent manner, as assessed by p70S6K1 levels (S1D–S1F Fig). However, p70S6K1 levels remained unchanged in SesnTKO or Nprl13KO cells treated with NV-5297 or leucine, demonstrating only basal mTORC1 activity [46]. Thus, an intact Sestrin/GATOR pathway is required for NV-5297 activity (S1D–S1F Fig).

Next, we assessed NV-5297 *in vivo* in C57BL/6 mice. C57BL/6 mice were treated orally with NV-5297 (160 mg/kg) and pharmacokinetics measured. The compound half-life of NV-5297 was approximately 3 hours, the time to maximal concentration ($t_{max}$) was 30 minutes, and the oral bioavailability was 100% (S1 Table). NV-5297 functionality was assessed by measuring phosphorylated ribosomal protein S6 [pS6], a direct target of S6K after oral administration of NV-5297 or vehicle. NV-5297-treated mice had increased pS6 relative to vehicle-treated animals in peripheral tissues collected 1 hour later (Fig 1B and 1C). In the brain, NV-5297-treatment increased pS6 in all regions sampled relative to controls (Fig 1D and 1E).

To test NV-5297's potential as an HD therapeutic it was assessed in N171-82Q HD mice [41]. Hemizygous N171-82Q mice [HD mice] were dosed orally with NV-5297 (160 mg/kg) daily for one week, after which tissue pS6 levels were assessed. Notably, pS6 increased in NV-5297-treated mice striata (S2A and S2B Fig) and heart (S2A–S2C Fig) relative to vehicle-treated N171-82Q mice. We cannot discount that the pharmacokinetics of NV-5297 in the N171-82Q mice could be slightly different from the controls despite the normalization of the pS6 level. Moreover, NV-5297 treatment normalized the ratio of pS6 level in the striatum and heart to those of age-matched WT C57BL/6 mice (S2A–S2C Fig).

## Daily NV-5297 dosing ameliorates motor behavior and striatal deficits in HD mice

Progressive muscle loss in HD mice decreases strength [47, 48] and accelerates a decrease in rotarod motor performance [49]. Starting at 6 weeks of age, N171-82Q mice were treated daily with oral NV-5297 (160 mg/kg) or vehicle for 6 weeks. Forelimb grip strength and accelerating rotarod performance were measured at baseline (5 weeks old) and treatment endpoint (12 weeks old, Fig 2A). NV-5297 did not impact weight loss or forelimb grip strength [FGS], both of which decreased over time in all mice (S3A and S3B Fig). The performance component of the rotarod, as measured through the improvement of the task performance through learning and coordination over time, were similar for all groups at baseline [41]. However, at 12 weeks of age when a clear motor decline is expected, drug-treated animals showed improved rotarod motor performance while vehicle-treated animals did not (Fig 2B).

Striatal tissues collected at endpoint showed that pS6 and the striatal medium spiny neuron [MSN] marker DARPP-32 were significantly elevated in tissues from NV-5297-treated HD mice compared to controls, indicating mTORC1 activation and improved striatal health (Fig 3A and 3B). DARPP-32 is a specific marker of the medium spiny neurons (MSN) constituting 95% of the cell population within the striatum. DARPP-32 is a fundamental component of the

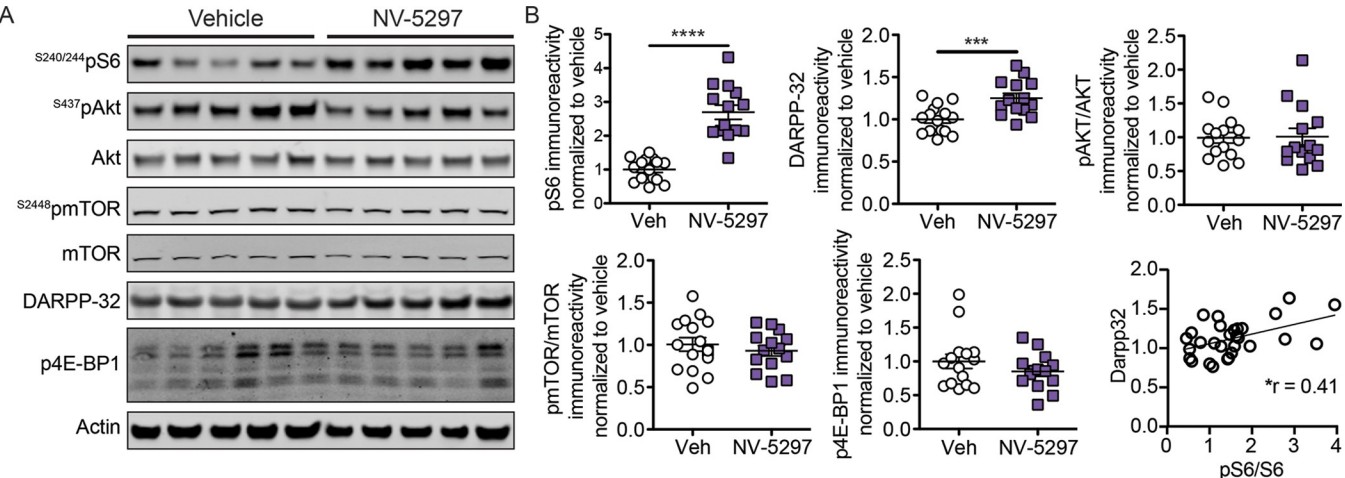

**Fig 3. Activation of the mTORC1 pathway in the striatum following daily oral dosing of NV-5297.** (A) Representative immunoblots of the striatum for normalized phosphorylated S6 [$^{S240/244}$pS6], S6, phosphorylated Akt [$^{S437}$pAkt], Akt, phosphorylated mTOR [$^{S2448}$pmTOR], mTOR, DARPP-32, phosphorylated 4E-BP1 [$^{Thr37/46}$p4E-BP1], 4E-BP1 and Actin from homogenized striatum isolated from N171-82Q males after 6 weeks of daily oral dosing of NV-5297 (160 mg/kg) or vehicle [Veh]. (B) Quantification of the immunoblots from homogenized striatum isolated 1 hour after the final oral dosing with NV-5297 (160 mg/kg; n = 14) normalized to loading control levels and further normalized to vehicle treated mice (n = 15). A significant Spearman correlation is present between pS6/S6 and DARPP32 levels. Data are presented as mean ± SEM. $^{*}P \leq 0.05$, $^{***}P \leq 0.001$, $^{****}P \leq 0.0001$ (one-tailed unpaired Student's t-test [pmTOR/mTOR, DARPP-32], with a Welch's correction [pS6/S6, p4E-BP1/4E-BP1], Mann-Whitney test [pAKT/Akt].

dopamine-signaling cascade, and its expression is essential to the ability of dopamine to regulate the physiology of striatal neurons. Decreased DARPP-32 expression in HD starts very early in the disease and correlates with decreased neuron numbers, specifically in the striatum [50, 51] and has been used in several studies including by our group [38, 52]. The pS6/S6 ratio was also positively correlated to the increase in DARPP-32. Additionally, the ratio of phosphorylated to total protein kinase B [pAkt/Akt], pmTOR/mTOR and p4E-BP1/4E-BP1 assessed, and all were unchanged. The BCAA-sensing pathway of mTORC1 activation is based on subunit localization and an independent arm of the PI3K/Akt kinase cascade, so the lack of changes in pAkt and pmTOR is expected with NV-5297 treatment. On the other hand, 4E-BP1 phosphorylation is dynamic and has a delayed short-term inhibition. The lack of detectable changes could thus reflect both the short half-life of NV-5297 and the collection time-point chosen [53]. Additionally, S6 is an indirect mTORC1 target and thus benefits from intracellular signal amplification, making it therefore a more sensitive surrogate of mTORC1 activity particularly in tissues. The increased mTORC1 activity identified through pS6 may therefore not pass the high threshold needed to detect differences in 4E-BP1 phosphorylation.

Consistent with elevated DARPP-32 levels, NV-5297-treated mice showed significant increases in striatal volume relative to vehicle-treated mice (Fig 4A and 4B), but no change in medium spiny neuron area (Fig 4A and 4C), suggesting that this difference can be mainly attributed to neuronal death within the striatum. These findings partially recapitulate those seen with viral-mediated mTORC1 activation [38].

## NV-5297 treatment ameliorates heart function in N171-82Q mice

Multiple HD mouse models demonstrate baseline cardiac fibrosis that becomes exacerbated with stress [15–17, 19–21]. Chronic mTORC1 stimulation can partially rescue these phenotypes [19]. To test NV-5297 on cardiac phenotypes in HD mice, N171-82Q animals were dosed daily with drug or vehicle for six weeks and heart function evaluated by transthoracic echocardiography at multiple timepoints (Fig 2A). NV-5297 treatment increased the ejection

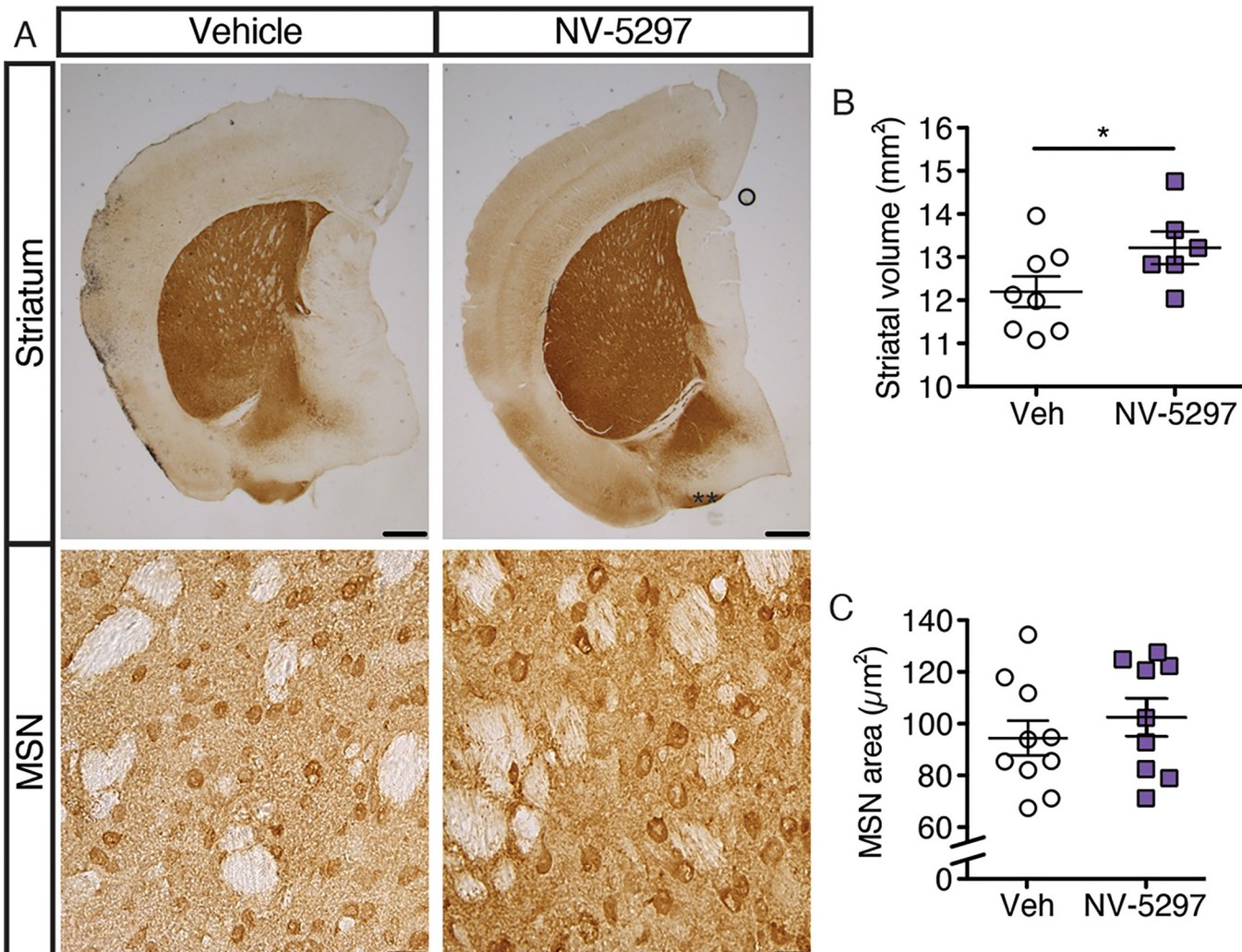

**Fig 4. Daily oral dosing of NV-5297 over 6 weeks protects the striatum in N171-82Q males.** (A) Representative immunohistochemical staining of Medium Spiny Neurons (MSN) with anti-DARPP-32 in 12 weeks old N171-82Q males. Scale bars striatum: 500 μm. Scale bars MSN: 20 μm. Quantification of striatal volumes (n = 8 Vehicle [Veh], n = 6 NV-5297) (B) and MSN cell area (Veh n = 9, NV-5297 n = 9) (C). Data are presented as mean ± SEM. $^*P \leq 0.05$ (unpaired one-tailed Student's t-tests).

fraction and fractional shortening when compared to the vehicle-treated animals; there was, however, no discernable effect on heart mass with drug treatment (Fig 5A–5C). Post-necropsy assessment of heart tissues showed elevated pmTOR, pS6 and p4E-BP1 levels in NV-5297-treated mice versus controls (Fig 5D and 5E).

## NV-5297 treatments improve adaptive cardiac hypertrophy in HD mice

We next tested if daily NV-5297 dosing in HD mice could enable cardiac adaptation to chronic stress [19]. In this experiment, the NV-Sal and Veh-Sal groups have an osmotic pump implant on their back. The presence of this pump makes the echocardiography more challenging, justifying the difference to the baseline groups measurements along with the smaller sample size for this experiment. For this study, we found no significant interaction of stress and cardiac stress and therefore we focused on the changes within the vehicle-treated groups and within the NV-5297-treated groups. Six-week old N171-82Q male mice with similar baseline rotarod

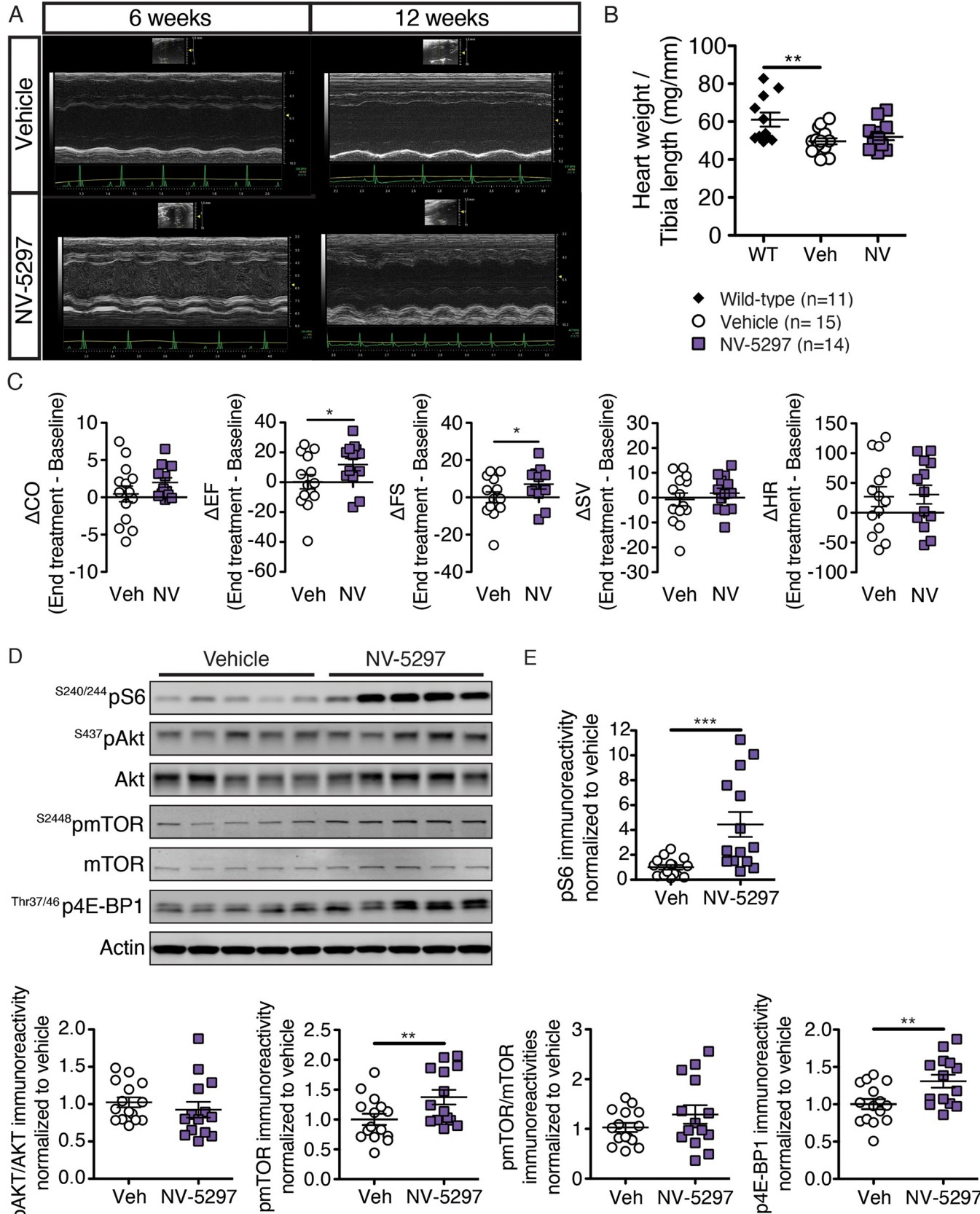

**Fig 5. Oral dosing with NV-5297 over 6 weeks improves cardiac function and activates the mTORC1 pathway.** (A) Representative M-mode echocardiograms from transthoracic echocardiography at 6 weeks and 12 weeks of vehicle and NV-5297 treated N171-82Q males. (B) Heart weight corrected for the animal's standard length (tibia length) at 12 weeks. (C) Left ventricular measurements obtained by transthoracic echocardiography by deducting the end treatment (12 weeks) from the baseline (6 weeks) measurements. Variation in cardiac output [CO], ejection fraction [EF], fractional shortening [FS], stroke volume (SV) and Heart rate [HR]. (D) Representative immunoblots for normalized phosphorylated S6 (S240/244pS6), S6, phosphorylated Akt (S437pAkt), Akt, phosphorylated mTOR (S2448pmTOR), mTOR, phosphorylated 4E-BP1 (Thr37/46p4E-BP1), 4E-BP1 and Actin in homogenized heart. (E) Quantification of the immunoblots for homogenized heart isolated 1 hour after oral dosing with NV-5297 (160 mg/kg; n = 14) normalized to loading control levels and further normalized to vehicle (Veh) treated mice (n = 15). Data are presented as mean ± SEM. $^*P \leq 0.05$, $^{**}P \leq 0.01$, $^{***}P \leq 0.001$ (unpaired Kruskal-Wallis test with Dunn's multiple comparison test [B], unpaired one-tailed Student's t-tests [C, E pmTOR] with a Welch's correction [pS6/S6, pmTOR/mTOR] or a Mann-Whitney test [pAkt/Akt, p4E-BP1/4E-BP1]).

and forelimb grip strength performance were randomized (S4A and S4B Fig) and given NV-5297 (160 mg/kg) or vehicle for 1 week, and then subjected to a baseline echocardiography. After 3 additional weeks of NV-5297 or vehicle dosing, osmotic pumps were implanted that contained saline [Sal] or isoprenaline [Iso], a β-adrenergic agonist that induces cardiac stress by chronically increasing heart rate, contractility and peripheral vasodilation [54] (Fig 6A).

As expected, the implant alone did not impact HD mice mortality [19], with 100% survival in the Veh-Sal and NV-Sal groups. However, in Iso-stressed HD mice dosed orally with vehicle, there was 62.5% mortality. This was reduced to 45% with NV-5297-dosing (Fig 6B), supporting improved survival with NV-5297 treatment in the setting of cardiac-induced stress.

Iso-induced cardiac stress led to a significant weight loss in vehicle-treated animals at both 1- and 2-weeks post-implantation compared to their weight at baseline. Conversely, the NV-5297-treated HD mice maintained their weight over the trial duration (Fig 6C). Additionally, vehicle-treated HD mice under cardiac stress had an increase in heart rate from baseline permitting an increase in cardiac output despite a decrease in stroke volume (Fig 6E). Conversely, NV-5297-treated HD mice under cardiac stress experienced improved cardiac function through increased ejection fraction, fractional shortening and stroke volume from baseline leading to the increase in cardiac output (Fig 6D and 6E). Heart mass is also elevated in NV-5297-treated animals subjected to cardiac stress (Fig 6F). These modifications were observed by echocardiography as early as one week after the cardiac stress onset (S4C Fig). The improved heart function in response to cardiovascular stress in NV-5297-treated mice are consistent with the drug permitting the appropriate adaptive response otherwise absent in HD mouse hearts [19]. CO is calculated as the heart rate multiplied by the SV (the blood volume ejected from the left ventricle per beat). Both Veh-Iso animals and NV-Iso animals have an increase in CO compared to baseline but through different mechanisms. The increase in CO in Veh-Iso animals is due to the increase in heart rate observed to compensate for the decreased SV observed. This decrease in SV likely due to the increase in fibrosis we observed as fibrotic tissue is stiffer and reduces muscle contractility. On the contrary, the NV-Iso animals maintain a normal heart rate but increase their CO through a more efficient contraction that led to an increase in the SV. This increase in SV is achieved through the increase in the FS which measures the contraction as the muscle during the contraction (ventricle diameter at the end of the diastole–ventricle diameter at the end of the systole). In turn, this increase in FS lead to an increase in EF (percentage of blood ejected by the left ventricle by heartbeat). These changes can be observed visually with the echocardiogram in Fig 6D. Taken together, the CO increase in NV-Iso animals is done in a sustainable manner (physiologic hypertrophy), with the cardiac muscle growing to accommodate the increase in workload and sustain the CO while the Veh-Iso animals cope with the cardiac stress with adverse impacts (pathologic hypertrophy) such as the fibrosis, severe weight loss and increase in the mortality rate.

At trial endpoint, cardiomyocyte size and the impact of treatment on mTORC1 activation was assessed. While the cardiomyocyte area increased in both drug-treated and vehicle-treated

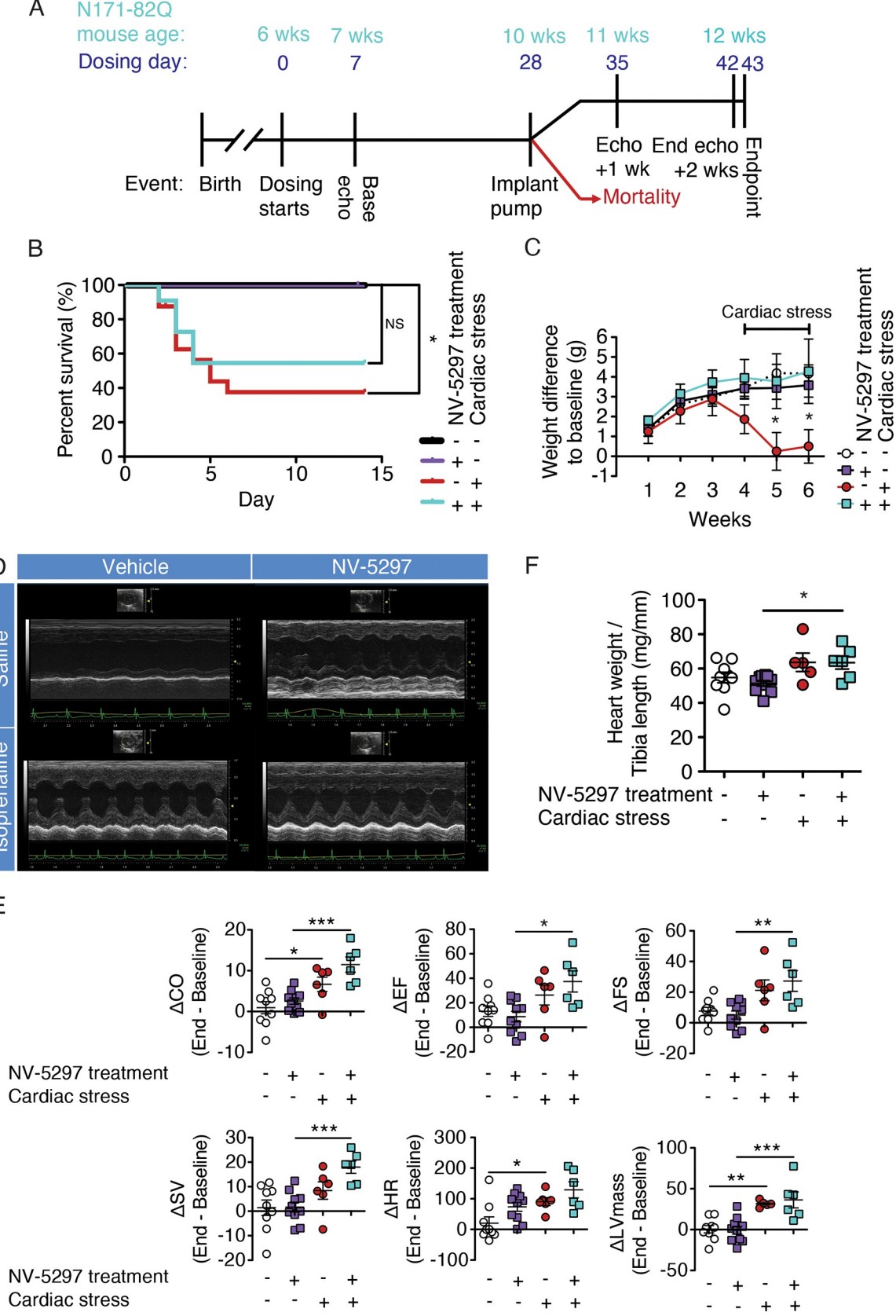

**Fig 6. NV-5297 oral dosing over 6 weeks improves the cardiac function and survival during cardiac stress.** (A) Experimental timeline. (B) Mortality of N171-82Q males in response to the cardiac stress (isoprenaline) or saline treatment for 14 days (Veh-Sal n = 9, NV-Sal n = 10, Veh-Iso n = 16, NV-Iso n = 11). (C) Body weight difference from baseline throughout the oral dosing and cardiac stress treatments. (D) Representative M-mode echocardiograms from transthoracic echocardiography at 12 weeks after 2 weeks of saline (Sal) or isoprenaline (Iso) exposure in vehicle (Veh) and NV-5297 (NV) treated N171-82Q males. (E) Left ventricular measurements obtained by transthoracic echocardiography by deducting the end treatment (12 weeks) from the baseline (6 weeks) measurements. CO, cardiac output; EF, ejection fraction; FS, fractional shortening; HR: Heart rate; LVmass: Left ventricular mass; SV: stroke volume. (F) Final heart weight corrected for the animal's standard length (tibia length). (Veh-Sal n = 9, NV-Sal n = 10, Veh-Iso n = 6, NV-Iso n = 6 in C-F). Data are presented as mean ± SEM. $^{*}P < 0.05$, $^{**}P < 0.01$, $^{***}P < 0.001$ (Mantel-Cox test [B], 3-way ANOVA [C], Mixed-effect model with Sidak's multiple comparisons test [C-F]).

mice exposed to chronic cardiovascular stress, fibrosis was present solely in Iso-infused HD mice treated with vehicle; hearts from Iso-stressed mice treated with NV-5297 remained devoid of fibrosis (Fig 7A–7D). Under cardiac stress, NV-5297-treated HD mice experienced

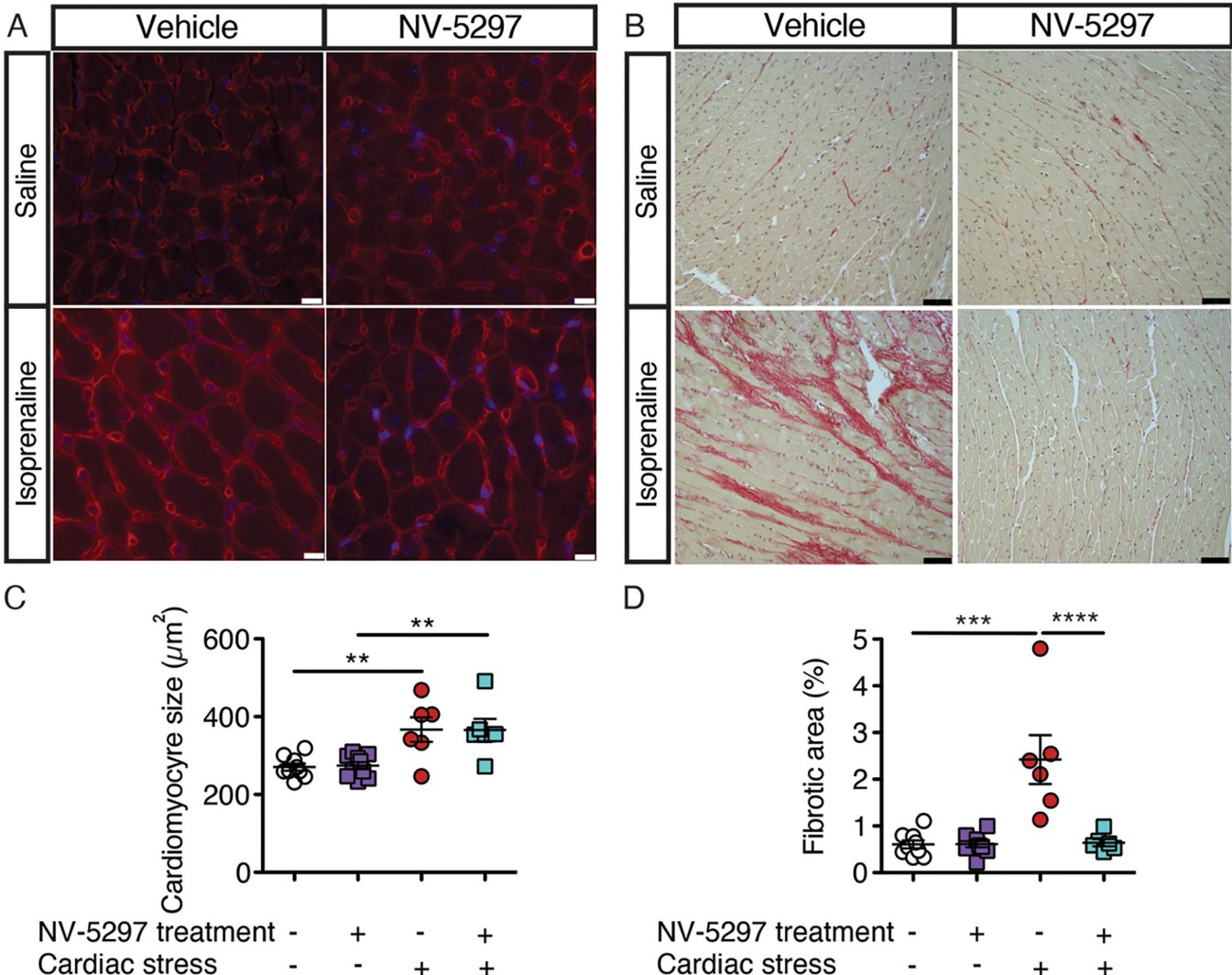

**Fig 7. Daily oral dosing of NV-5297 protects against cardiac fibrosis during cardiac hypertrophy following cardiac stress.** (A) WGA staining of cardiomyocytes. (B) Picrosirius red staining for collagen fibers. (C) Cardiomyocytes cross-sectional area measurement. Scale bars, 10 μm. (D) Quantification of collagen staining in the left ventricle of the heart. Scale bars, 50 μm. (Vehicle-Saline [Veh-Sal] n = 9, NV-5297-Saline [NV-Sal] n = 10, Vehicle-Isoprenaline [Veh-Iso] n = 6, NV-5297-Isoprenaline [NV-Iso] n = 6). Data are presented as mean ± SEM. $^{**}P \leq 0.01$, $^{***}P \leq 0.001$, $^{****}P \leq 0.0001$ (Mixed-effect model with Sidak's multiple comparisons tests).

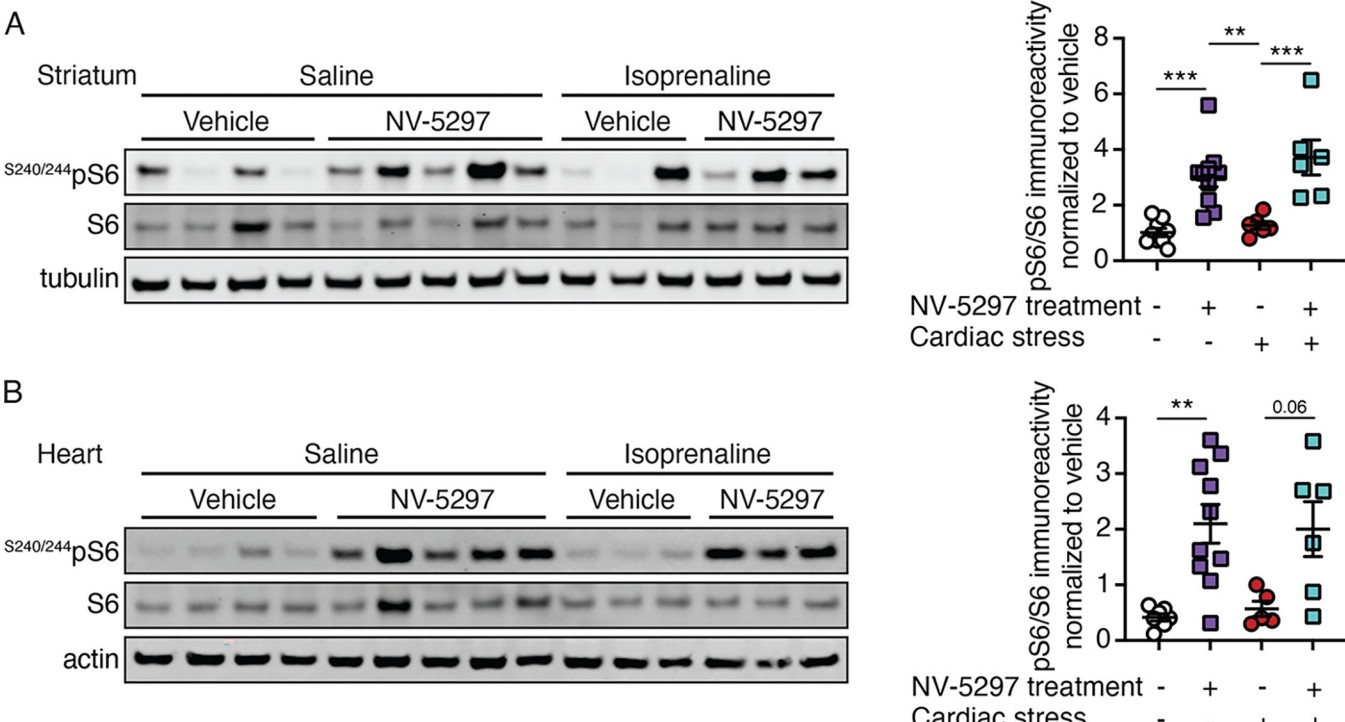

**Fig 8. mTORC1 pathway activation following NV-5297 dosing and cardiac stress in the striatum but not heart.** Representative immunoblot (left panels) of phosphorylated S6 ($^{S240/244}$pS6) and quantification of normalized level (right panels) for homogenized heart (A) and striatum (B) isolated 1 hour after oral dosing with NV-5297 (160 mg/kg) with a saline (n = 10) or isoprenaline (n = 6) implant or treated with vehicle and an isoprenaline implant (n = 6) normalized to loading control levels and further normalized to vehicle treated mice with a saline implant (n = 9). Data are presented as mean ± SEM. $^*P \leq 0.05$, $^{**}P \leq 0.01$, $^{***}P \leq 0.001$ (Mixed-effects model with Sidak's multiple comparisons test).

an increased pS6/S6 ratio in the striatum and a strong trend toward increased ratios in the heart (Fig 8A and 8B). These data indicate improved cardiomyocyte viability as a result of NV-5297 treatment, likely resulting from increased physiologic reserve and adaptation.

## Discussion

The studies reported here support the use of NV-2897 as a potential therapeutic agent for treatment of HD. Among the many HD cellular phenotypes, mTORC1 activity is decreased at baseline in HD brains, a finding recapitulated in animal models. While mTORC1 inhibitors are widely used, there are currently no small molecule mTORC1 activators available for clinical use. NV-5297 is thus a promising agent to correct a critical cellular pathway implicated in HD.

*In vitro* NV-5297 activity mirrors that of NV-5138, a closely related compound currently being tested clinically for treatement of depression. Both are orally bioavailable and activate mTORC1 through the leucine-sensing arm, and both have biologic activity *in vivo*. Here, we additionally find that NV-5297 ameliorates HD phenotypes in mice. These results are consistent with earlier approaches using virus-mediated constitutive expression of Rheb, the activating subunit of the mTORC1 complex [19, 38]. Importantly, and despite some individual variability in the response to the treatment, we show that transient mTORC1 pathway activation with NV-5297 is sufficient to improve striatal and cardiac health in HD mice.

mTORC1 integrates a wide array of environmental signals and produces an enormous range of downstream effects. Previous work investigating how mHTT expression results in mTORC1 dysregulation failed to identify a dominant pathway, though mislocalization of

individual subunits may have a role in the mechanism [19]. Similar investigation of individual downstream mechanisms failed to implicate one more than another in HD pathogenesis [19, 38]. Consequently, decreased mTORC1 activity likely contributes to the HD phenotype through its nonspecific functions promoting global protein translation, cellular growth, and anabolic reactions. The internal metabolic state is thus shifted towards catabolism and preservation, even when nutrients are readily available. Treatment with NV-5297 compensates for the HD-related dysregulation and restores the cell's ability to properly balance its metabolic state with the external environment in multiple tissues.

Maintaining the mTORC1 homeostatic level is critical for neuronal health [30], and mTORC1 pathway dysregulation is upstream of various phenotypic changes associated with HD. Rhes, the mTOR activating subunit, is highly expressed in the striatum and modulates the sensitivity of dopamine D1 and D2 receptors, which are highly expressed in the MSNs. Thus, the mTORC1 pathway is mechanistically linked to motor coordination deficits, especially on the rotarod, a task very sensitive to striatum integrity [30]. Rotarod motor performance improvement over consecutive days, integrating both coordination and motor learning, is especially dependent on striatal contribution [55]. In these studies, NV-5297-treated HD animals demonstrated improved rotarod performance over four consecutive days, whereas vehicle-treated mice had no change in their performance. While both groups had decreased performance relative to their presymptomatic and pre-trial baseline, the performance improvement observed in NV-5297-treated mice suggests a protective effect on striatal health in the setting of HD.

NV-5297 treatment increased striatal mTORC1 activation indicated by S6 phosphorylation, demonstrating the drug's ability to act on central nervous system tissues. Furthermore, NV-5297 treatment ameliorated MSN pathology in HD mice brains compared to vehicle-treated controls, as evidenced by increased DARPP-32 levels and greater striatal volume without MSN cell body enlargement. Like other HD fragment models, the primary neuropathologic phenotype in N171-82Q mice loss of MSN neurites rather than neuronal death [41]. Because DARPP-32 expression in the striatum is specific to MSNs, the combination of biochemical and histologic findings in these studies suggests NV-5297 treatment in HD mice is associated with increased MSN neurite density and is thus protective in the setting of mHTT expression. MSN neurite density should be assessed in a follow-up study to confirm this hypothesis.

The neurologic focus of these studies was the striatum because of MSN selective vulnerability in HD and its profound impact on phenotype. Behavioral, histologic, and biochemical analyses indicated that NV-5297 treatment improved striatal health in HD mouse models. However, other brain regions including deep cortical neurons are also affected in HD. Previous evaluation of the related compound NV-5138 found improved synaptic formation in the medial prefrontal cortex [40]; similar studies on NV-5297 in HD models could help elucidate how mTORC1 dysregulation in non-striatal brain regions contributes to disease phenotype.

The mTORC1 pathway promotes heart growth by enhancing protein translation via phosphorylated S6 and 4E-BP1 [56]. Murine cardiomyocytes reach their adult size around 3 months of age (approximately 12 weeks) [57]. NV-5297 treatment was initiated during the period of heart growth and did not significantly increase heart size. It is possible that initiating treatment earlier or using a higher dose may result in heart sizes closer to WT, though other systemic factors and molecular pathways could also be implicated [58]. Regardless, previous work has found that small heart size is not associated with functional abnormalities in this model; furthermore, constitutive mTORC1 activation through genetic modalities likewise did not affect adult heart size [19]. Hence, the small size may not have physiologic relevance and instead merely reflect the nutritional state of the animal. Nevertheless, we detected increased ejection fraction and fractional shortening following NV-5297 treatment, indicating enhanced

heart contractility [59]. The NV-5297 treatment, therefore, may be beneficial to HD heart physiology at baseline.

Cardiomyocyte growth can occur in either adaptive or pathological settings. Adaptation (also called physiologic hypertrophy) is protective and reduces the amount of work the heart has to exert in response to an increased workload in order to maintain the output [60, 61]. Normal contractility, structure, and function is preserved by increasing the cardiac mass and chamber volume in an organized and regulated manner [61, 62]. Cardiomyocyte size and myocardial mass increase through the addition of contractile proteins, a process that relies on the increased rate of protein synthesis mediated through the mTORC1 pathway activation [35, 60]. Conversely, chronic stress leads to a maladaptive response often termed pathologic hypertrophy [63]. Pathological hypertrophy is disordered with abnormal metabolic, structural, and functional properties, consequently leading to cardiomyocyte loss, inflammation, fibrosis, and declining contractility [64].

The mTORC1 pathway mediates adaptation in an organized manner while protecting the heart against the maladpative effects of hypertrophy. When inducing heart stress, S6 is initially increased and mediates an increased rate of protein synthesis [34, 35]. Akt, an upstream regulator of multiple pathways including mTORC1, is a pivotal regulator of hypertrophy and facilitates increased cardiac output, left ventricular mass, and chamber dilatation [36]. Overactivation of cardiac mTORC1 in a mouse model of hypertrophy protected the cardiomyocytes against interstitial fibrosis and inflammation; conversely, rapamycin, an mTORC1 pathway inhibitor, suppressed hypertrophy and accelerated the onset of heart failure [34]. Previous work in HD mice models found that hearts were unable to adapt in response to stress, but transduction with constitutively active Rheb restored the adaptive response and prevented mortality and cardiac fibrosis [19]. Here, we show that mTORC1 pathway activation by NV-5297 in HD mice had similar protective effects as genetic modification and promoted cardiac adaptation to stress. HD mice under isoprenaline-induced cardiac stress and treated with NV-5297 maintained efficient contractility throughout the heart stress period with no associated weight loss and survived longer than vehicle-treated treatment mice. NV-5297 treatement was associated with increased heart size and contractility (EF, FS) without fibrosis in response to stress, as opposed to the diffuse fibrotic response observed in vehicle-treated mice. Taken together, NV-5297 treatment promoted cardiac adaption in response to stress, whereas vehicle-treated HD mice showed signs of chronic heart disease, severe weight loss and decreased survival.

In conclusion, we show that NV-5297 successfully increases mTORC1 activity in CNS and peripheral tissues of HD mouse models and may be useful as a systemic therapy for HD. While optimization of dosing, timing, and duration will be essential for proper therapeutic application in HD patients, these experiments demonstrate proof-of-concept for the usage of a small molecule mTORC1 activator in treating multi-system HD phenotype. While the ideal therapeutic approach would be to eliminate the underlying mHTT, current technologies have either failed to demonstrate efficacy in clinical trials and/or are limited in their biologic distribution. NV-5297 is thus a promising modality to complement targeted HD therapeutics with beneficial effects in both central and peripheral tissues.

## Supporting information

**S1 Fig. NV-5297 activates mTORC1 by modulating the interaction between Sestrin2 and GATOR2.** (A) Activation of the mTORC1 pathway by NV-5297. Immunoblot shows levels of phosphorylated S6K1 ($^{T389}$pS6K1) and non-phosphorylated S6K1 in HEK-293T cells starved of leucine (Leu) for 50 minutes followed by addition of vehicle (Veh), NV-5297 or leucine for

10 minutes and the associated quantification. (B) NV-5297 disrupts Sestrin2/GATOR2 in the *in vitro* Sesn2/WDR24 protein-protein interaction assay in a dose-dependent manner. Immunoblotting of Sestrin 2 (Sesn2) after flag-WDR24 immunoprecipitation from amino acid-starved flag-WDR24 expressing HEK-293T cells followed by addition of NV-5297 or leucine at 10 μM for 10 minutes. (C) Dose-dependent activation of mTORC1 by NV-5297 or leucine correlates with disruption of Sesn2 from Flag-WDR24. Immunoblot shows levels of Sestrin2 bound to immunoprecipitated Flag-WDR24 and levels of $^{T389}$pS6K1 in flag-WDR24 expressing HEK-293T cells were starved of leucine for 50 minutes followed by addition of vehicle, NV-5297 or leucine for 10 minutes. NV-5297 requires an intact Sestrins/GATOR pathway to mediate mTORC1 pathway activation. Representative immunoblots of $^{T389}$pS6K1 and quantification of the normalized level of $^{T389}$pS6K1 from 8 independent experiments (F) show TORC1 activity in unedited HEK-293T cells, HEK-293T cells deficient for Sestrin 1 (Sesn1), 2 (Sesn2) and 3 (Sesn3) (D) or HEK-293T cells deficient for the GATOR1 component Nprl3 (E) after 50 minutes leucine starvation followed by the addition of vehicle, NV-5297 or leucine for 10 minutes. Data are presented as mean ± SEM. $^*P \leq 0.05$, $^{**}P \leq 0.01$ and $^{***}P \leq 0.001$ (One-way ANOVA with Tukey's multiple comparison tests).
(TIFF)

**S2 Fig. Activation of the mTORC1 pathway in mice striatum and heart after oral dosing with NV-5297.** (A) Representative immunoblot for the phosphorylated ($^{S240/244}$pS6), S6 and actin or tubulin in the striatum and heart of N171-82Q wild-type (WT) mice or N171-82Q HD mice respectively collected 1 hour after the last dose of a week of oral dosing with vehicle (Veh) or NV-5297 (160 mg/kg; A). Quantification of levels of $^{S240/244}$pS6 in the striatum (B) and the heart (C) normalized to loading control levels and further normalized to untreated WT mice. Data are presented as mean ± SEM. $^{**}P \leq 0.01$, $^{***}P \leq 0.001$ and $^{****}P \leq 0.0001$ (Kruskal-Wallis test with Dunn's multiple comparison tests).
(TIFF)

**S3 Fig. Oral dosing of NV-5297 for 6 weeks does not alter weight or forelimb grip strength.** (A) Weight remained unaltered between N171-82Q male mice orally dosed with NV-5297 (160 mg/kg) for 6 weeks and vehicle treated males. (B) Forelimb grip strength also remained unaltered after 6 weeks of NV-5297 or vehicle-treated dosing. The forelimb grip strength decreased with time similarly in both groups. Data are presented as mean ± SEM. $^{**}P \leq 0.01$ (two-way repeated measure ANOVA followed by Tukey's multiple comparisons test [A] and Dunnett's multiple comparison tests [B]).
(TIFF)

**S4 Fig. A week of cardiac stress modifies the N171-82Q male mice cardiac function.** The baseline latency to fall from the accelerated rotarod (A) and the forelimb grip strength (B) was similar between the experimental groups. (C) N171-82Q male mice orally dosed daily with NV-5297 (160 mg/kg) or vehicle [Veh] starting at 6 weeks of age showed modifications in their heart function a week after implantation (at 10 weeks of age) of a saline [Sal] or isoprenaline [Iso] osmotic pump. CO: Cardiac output, EF: Ejection fraction, FS: Fractional shortening, SV: Stroke volume, HR: heart rate, LVmass: Left ventricular mass. (Veh-Sal n = 9, NV-Sal n = 10, Veh-Iso n = 6, NV-Iso n = 6). Data are presented as mean ± SEM. $^*P \leq 0.05$, $^{**}P \leq 0.01$ (Mixed-effects models [A, B] with Sidak's multiple comparisons tests [C]).
(TIFF)

**S1 Table. Pharmacokinetics parameters of NV-5297 after an intravenous or oral dose in C57BL/6 mice.** CL: Clearance, F: Bioavailability, INF: Infinity, MRT: Mean residence time, OD: Oral dose, PK: Pharmacokinetics; $T_{max}$: time to reach $C_{max}$, $V_{ss}$: Steady state volume of

distribution.
(PDF)

**S1 File. Synthesis of NV-5297 [(S)-2-amino-5,5,5-trifluoro-4,4-dimethylpentanoic acid].**
(PDF)

**S1 Raw images.**
(PDF)

## Acknowledgments

We thank Amy Muehlmatt for help with the oral gavage of animals. We thank Ellie Carrell for her valuable editorial suggestions. We thank the CHOP Pathology core for preparing the tissue for histology.

## Author Contributions

**Conceptualization:** Sophie St-Cyr, Daniel D. Child, Beverly L. Davidson.

**Data curation:** Sophie St-Cyr, Daniel D. Child, Emilie Giaime, Beverly L. Davidson.

**Formal analysis:** Sophie St-Cyr, Daniel D. Child, Seung Hahm, Beverly L. Davidson.

**Investigation:** Daniel D. Child, Emilie Giaime, Alicia R. Smith, Christine J. Pascua.

**Methodology:** Sophie St-Cyr, Daniel D. Child, Christine J. Pascua, Beverly L. Davidson.

**Project administration:** Beverly L. Davidson.

**Supervision:** Sophie St-Cyr, Daniel D. Child, Eddine Saiah, Beverly L. Davidson.

**Validation:** Sophie St-Cyr.

**Writing – original draft:** Sophie St-Cyr, Daniel D. Child.

**Writing – review & editing:** Sophie St-Cyr, Daniel D. Child, Eddine Saiah, Beverly L. Davidson.

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
