## [Decision Letter · Decision Letter 0]

10 Jul 2022

PONE-D-22-11888Huntington’s disease phenotypes are improved via mTORC1 modulation by small molecule therapyPLOS ONE

Dear Dr. Davidson,

Thank you for submitting your manuscript to PLOS ONE. After careful consideration, we feel that it has merit but does not fully meet PLOS ONE’s publication criteria as it currently stands. Therefore, we invite you to submit a revised version of the manuscript that addresses the points raised during the review process. I have received two completed reviews. Both of the reviewers thought the manuscript contains very interesting results but have doubts in experimental design, data interpretation and statistical analysis. I believe the manuscript can be significantly improved if these questions can be addressed.

We look forward to receiving your revised manuscript.

Kind regards,

Yuqing Li, Ph.D.

Academic Editor

PLOS ONE

Journal Requirements:

“BLD received sponsored research support from Navitor pharmaceuticals.  EG, SH, and ES are employees of Navitor Pharmaceuticals”

“BLD received sponsored research support from Navitor pharmaceuticals.  EG, SH, and ES are employees of Navitor Pharmaceuticals”

7. Please upload a copy of Figure 6 and 7, to which you refer in your text on page 22, 23 and 24. If the figure is no longer to be included as part of the submission please remove all reference to it within the text.

Additional Editor Comments:

I have received two completed reviews. Both of the reviewers thought the manuscript contains very interesting results but have doubts in experimental design, data interpretation and statistical analysis. I believe the manuscript can be significantly improved if these questions can be addressed.

Reviewers' comments:

Reviewer's Responses to Questions

**Comments to the Author**

1. Is the manuscript technically sound, and do the data support the conclusions?

Reviewer #1: Partly

Reviewer #2: Partly

2. Has the statistical analysis been performed appropriately and rigorously? 

Reviewer #1: I Don't Know

Reviewer #2: Yes

3. Have the authors made all data underlying the findings in their manuscript fully available?

Reviewer #1: Yes

Reviewer #2: Yes

4. Is the manuscript presented in an intelligible fashion and written in standard English?

Reviewer #1: Yes

Reviewer #2: Yes

5. Review Comments to the Author

Reviewer #1: St-Cyr et al report on effect of an mTORC1 pathway activator small molecule, NV-5297, on cardiac and striatal measures in the N171-82Q mouse. The study reported on PK-PD effects of the compound on mTORC1 pathway activation by following pS6 S240/242 phosphorylation along with additional biochemical and histological measures in striatum and heart. Effects of treatment on brain were assessed by measuring rotarod, MSN area and striatal volume. Heart outcomes were measured using echocardiograms on mice under baseline or stress conditions.

The manuscript is well organized and well written. The methods applied are rigorous and the authors do a nice job linking biochemical effects of the treatment to striatal and cardiac outcome measures in the N171-82Q mouse.

Overall, I am concerned that many of the significant effects reported (see below) were not robust and some may have been driven by some outlier animals. The authors should try to address these specific comments. The effects of the NV-5297 treatment appeared to be more robust in heart compared to brain (striatum); and the authors argue the importance of cardiac involvement in HD. There are numerous publications reporting cardiac abnormalities in HD mouse models, but the clinical evidence of cardiac involvement in HD is not fully understood and certainly not well accepted by HD researchers and clinicians. I suggest some effort is taken to recognizing this gap and putting in context the translatability of NV-5287 for HD, given the focus of this manuscript on improvement of cardiac measures in the mouse model.

Specific Comments:

1. There are various typos throughout the text. Please re-run spell-check.

2. FIG 2 Rotarod. Is significant effect of treatment shown by ANOVA or is data shown only significant – post hoc t test on Day 4 of the test run at 12 weeks?

3. FIG 3 DARPP32. Is there evidence of a genotype specific – WT vs N171-82 mouse decrease in DARPP32 @ 12 weeks. ie. Are DARPP32 levels returned to WT baseline?

In FIG3, there appears to be some variability in pS6/S6 response due to treatment in striatum – with about half the animals appearing not to have any change vs vehicle. A bit surprised this resulted in p<0.001 significance given the high degree of overlap of the data points. Same appears true for the DARPP32 data. Suggest authors try correlating pS6/S6 and DARPP32 levels in individual animals. ie. Scatter plot.

4. FIG 4. Striatal volume. Remove phrase on page 20, last paragraph – that MSN area is showing an “upward trend”. Not significantly valid and not at all convincing from the data. It appears from methods section that multiple striatal sections from each animal were taken and data represented in FIG 4A is quantification down the Z stack. Is that correct and if so, can you clarify about how many sections per animals were analysed? Also, did you try quantifying DARPP32 cellular intensity (MSNs?) Appears to be increased in FIG 4B image.

5. FIG 5C. Cardiac measures. EF and FS reported as improved by treatment. The significance effect appears to be driven by 1 (low) outlier in vehicle group and 1 (high) outlier in treatment group – with the balance of 14 animals appearing to be not changed between vehicle and treatment. Again, I am surprised this reaches significance and it is not at all convincing there is a biological effect here.

6. FIG 5D. pS6 immunoreactivity. Again, it appears there are responder and non-responders amongst all animals tested. Please expand on this in your discussion of the results and what that may mean. Same comment for pmTor and p4E-BP1 data.

7. FIG. 6. Cardiac function and survival. FIG 6B. Were animals monitored past 15 days post cardiac stress? Appears about half of the animals do not die acutely from the procedure. FIG 6E. Significant effects of treatment on EF, FS, SV and LVM (after cardiac stress) appear to be extremely subtle and significance appears to be driven by very few outlier animals. Please address.

Reviewer #2: The paper proposed NV-5297 as a promising molecule for treating phenotypes of Huntington’s Disease (HD). Using cell culture, the authors first approved NV-5297 activates mTORC1 pathway by disrupting the link between Sestrin2 and GATOR2. Then the drug was tested in a HD model, with increased motor control, striatal maker and volume, and cardiac functions observed. Overall, the paper is of interest for the development of HD therapeutics. However, there lacks an important control group of WTs to be compared with vehicle treated HD mice group and NV-5297 treated HD mice group. Below are some specific suggestions:

1. Line 361 and Figure S1, it mentioned several times that there is a dose-dependent function of NV-5297, which is not obvious in the figures. A panel of quantification or larger dose differences may be helpful in this case.

2. Line 393 and Figure 1, the authors accessed the pharmacokinetics and drug distribution in WTs. However, in later experiments, N171-82Q mice were used. How to guarantee the drug behaves the same in different mouse models?

3. Line 406, the label of the figure is wrong. Figure 2 does not contain the data about pS6 levels in the striatum and heart, neither does Supplementary figure 2, which only shows a quantification of pS6/S6.

4. Line 428 and Figure 3, the author measured the level of DARPP-32, but not the phosphorylated DARPP-32. Previous research shows that the function of DARPP-32 depends on phosphorylation and the phosphorylation sites. Plus, the function of DARPP-32 in striatum is controversial. Therefore, more evidence is required for the link between increased DARPP-32 and improved striatal health.

5. Line 452 and Figure 4, at least one more group of WTs with vehicle treatment is required.

6. Line 470 and Figure 5, in the main context, it says that there are elevations of pS6/S6 and p4E-BP1/4E-BP1. However, there is no corresponding plots in the figure. S6 and 4E-BP1 are also missed in the representative immunoblots.

7. Figure 6B, is the legend of Veh-Sal group purposely to be shorter than the others? It is hard to distinguish Veh-Sal and NV-Sal group in the figure (only Veh-Sal group is able to be seen).

8. Figure 6E, the results of Veh-Sal without cardiac stress and NV-Sal without cardiac stress is different from Figure 5. There are changes in EF and FS between NV-5297 treatment group compared with vehicle group in Figure 5, but not here. Secondly, is there any difference between the Veh-Sal group with cardiac stress and the NV-Sal group with cardiac stress? Since there is no WT group, it is hard to tell which group is closer to “normal”. Additionally, why the increases in cardiac output and heart rate are not treated as readout of increased cardiac function? In the introduction, line 56, it says that phenotypes like smaller heart size and impaired cardiac function are exacerbated by chronic ß-adrenergic agonist in HD mouse models. However, figure 6E shows increased cardiac output and heart rate in Veh-Sal group with cardiac stress. Are the results here contradicting to the previous statement?

9. Discussion Line 592, not enough evidence for the conclusion about increased MSN neurite density.

6. PLOS authors have the option to publish the peer review history of their article (what does this mean?). If published, this will include your full peer review and any attached files.

Reviewer #1: No

Reviewer #2: **Yes: **Shangru Lyu

---

## [Author Response · Author response to Decision Letter 0]

15 Jul 2022

Response to Reviewers

PONE-D-22-11888

Huntington’s disease phenotypes are improved via mTORC1 modulation by small molecule therapy

Reviewer #1: St-Cyr et al report on effect of an mTORC1 pathway activator small molecule, NV-5297, on cardiac and striatal measures in the N171-82Q mouse. The study reported on PK-PD effects of the compound on mTORC1 pathway activation by following pS6 S240/242 phosphorylation along with additional biochemical and histological measures in striatum and heart. Effects of treatment on brain were assessed by measuring rotarod, MSN area and striatal volume. Heart outcomes were measured using echocardiograms on mice under baseline or stress conditions.

The manuscript is well organized and well written. The methods applied are rigorous and the authors do a nice job linking biochemical effects of the treatment to striatal and cardiac outcome measures in the N171-82Q mouse.

Overall, I am concerned that many of the significant effects reported (see below) were not robust and some may have been driven by some outlier animals. The authors should try to address these specific comments. The effects of the NV-5297 treatment appeared to be more robust in heart compared to brain (striatum); and the authors argue the importance of cardiac involvement in HD. There are numerous publications reporting cardiac abnormalities in HD mouse models, but the clinical evidence of cardiac involvement in HD is not fully understood and certainly not well accepted by HD researchers and clinicians. I suggest some effort is taken to recognizing this gap and putting in context the translatability of NV-5287 for HD, given the focus of this manuscript on improvement of cardiac measures in the mouse model.

Specific Comments:

1. There are various typos throughout the text. Please re-run spell-check.

Answer 1-1: The spell-check was re-run.

2. FIG 2 Rotarod. Is significant effect of treatment shown by ANOVA or is data shown only significant – post hoc t test on Day 4 of the test run at 12 weeks?

Answer 1-2: The difference is within-group, meaning that NV-5297-treated HD mice performance improved over time (main effect with ANOVA) while there is no such improvement in vehicle-treated HD mice. The specific learning of the task, i.e. improvement of the performance over time, is under striatal control (Augustin et al. 2020 Neuropsychopharmacol, Shiotsuki et al. 2010 J Neurosci Methods). Thus, there is a significant effect of treatment suggesting striatal improvement.

3. A. FIG 3 DARPP32. Is there evidence of a genotype specific – WT vs N171-82 mouse decrease in DARPP32 @ 12 weeks. ie. Are DARPP32 levels returned to WT baseline?

B. In FIG3, there is variability in pS6/S6 response due to treatment in striatum – with about half the animals appearing not to have any change vs vehicle. A bit surprised this resulted in p<0.001 significance given the high degree of overlap of the data points. Same appears true for the DARPP32 data. Suggest authors try correlating pS6/S6 and DARPP32 levels in individual animals. ie. Scatter plot.

Answer 1-3: A. N172-82Q have reduced DARPP32 levels at 12 weeks of age compared to WT animals (Jang et al. 2018 Front Cell Neurosci). In that work, values were normalized to GAPDH, while we normalize to actin. Also, we use different blots negating a direct comparison. When evaluating the protein level ratios, the increase is roughly half of that presented in Jang et al. 2018. Note however that this increase is similar to our earlier observations (Lee et al. 2015, Neuron) at 10 weeks of age, where we normalized to actin in response to chronic Rheb activation. These data support our earlier findings, with a partial rescue as we describe.

B. The results are significant as stated. Further, we only report large effect size statistical results in this paper as described on line 303. In addition, we added a scatter plot of pS6/S6 to DARPP32 levels that better represents the significant Spearman correlation (panel in Figure 3B) (line 316, 432, in caption line 452).

4. FIG 4. Striatal volume. Remove phrase on page 20, last paragraph – that MSN area is showing an “upward trend”. Not significantly valid and not at all convincing from the data. It appears from methods section that multiple striatal sections from each animal were taken and data represented in FIG 4A is quantification down the Z stack. Is that correct and if so, can you clarify about how many sections per animals were analysed? Also, did you try quantifying DARPP32 cellular intensity (MSNs?) Appears to be increased in FIG 4B image.

Answer 1-4: As suggested, we changed the statement on line 458: �…but no change in medium spiny neuron area (Fig 4A and C). � We also added that 5 striatal sections were assessed per animal on line 269. We did not quantify DARPP32 cellular intensity as the protocol did not allow for an accurate way to control for the intensity of staining.

5. FIG 5C. Cardiac measures. EF and FS reported as improved by treatment. The significance effect appears to be driven by 1 (low) outlier in vehicle group and 1 (high) outlier in treatment group – with the balance of 14 animals appearing to be not changed between vehicle and treatment. Again, I am surprised this reaches significance and it is not at all convincing there is a biological effect here.

Answer 1-5: There are no outliers in the groups as measured by our standardized protocol of using ROUT in GraphPad Prism. Even if we remove all samples outside two standard deviations, these results remain significant. Again, we report only large effect sizes effect measurements (line 303). We expect variability in both groups for this type of measurement. Child et al. 2018 (Cell Reports) found a similar variability in echocardiography measurements, including in the WT mice group.

6. FIG 5D. pS6 immunoreactivity. Again, it appears there are responder and non-responders amongst all animals tested. Please expand on this in your discussion of the results and what that may mean. Same comment for pmTor and p4E-BP1 data.

Answer 1-6: It is difficult to say that we have non-responders as we cannot serially assess the mTORC1 pathway activation in striatal and heart tissue. We cannot discount the fact that the mTORC1 pathway activity could be potentially lower at study initiation, which the reviewers tags as ‘ non-responders’. No mice had explicit adverse effects to the treatment and the behavioral data were tight. Finally, the NV-5297 treatment consists of a mTORC1 pathway activation of ~ 3 hours per day compared to the constant activation tested in previous studies (Lee et al. Neuron, Child et al. 2018 Cell Rep), so we only ever expected a partial rescue of the HD phenotype. This transience nature justifies the multiple complementary measures that we report regarding striatal health (DARPP-32 level, striatal volume…) and cardiac function (contractility measurements, heart size, presence of fibrosis…) and that collectively indicate an improvement of the HD phenotype. We also now acknowledge this individual variability in the discussion (line 572).

7. FIG. 6. Cardiac function and survival. FIG 6B. Were animals monitored past 15 days post cardiac stress? Appears about half of the animals do not die acutely from the procedure. FIG 6E. Significant effects of treatment on EF, FS, SV and LVM (after cardiac stress) appear to be extremely subtle and significance appears to be driven by very few outlier animals. Please address.

Answer 1-7: FIG6B. Tissues were collected at Day 15 for analysis, so we did not continue the experiment. The surviving Veh-Iso animals were unhealthy by this point as indicated by their drastic weight loss (Fig 6C). Most of the echocardiography changes can be detected after 1 week (Suppl Fig 4). 

FIG6E. See answer 1-5 about outliers and large effect sizes. The sample sizes are small due to the 40% survival of the Veh-Iso group. The only result modified by more stringent outlier removal was LVM that we modified in the text (line 530) and figure 6E. However, the cumulative improvement of several contractility measures strengthens the finding of an improved contractility overall.

Reviewer #2: The paper proposed NV-5297 as a promising molecule for treating phenotypes of Huntington’s Disease (HD). Using cell culture, the authors first approved NV-5297 activates mTORC1 pathway by disrupting the link between Sestrin2 and GATOR2. Then the drug was tested in a HD model, with increased motor control, striatal maker and volume, and cardiac functions observed. Overall, the paper is of interest for the development of HD therapeutics. However, there lacks an important control group of WTs to be compared with vehicle treated HD mice group and NV-5297 treated HD mice group. Below are some specific suggestions:

1. Line 361 and Figure S1, it mentioned several times that there is a dose-dependent function of NV-5297, which is not obvious in the figures. A panel of quantification or larger dose differences may be helpful in this case.

Answer 2-1: A graph with the quantifications for the mTORC1 pathway activation by NV-5297 through S6K1 phosphorylation is now presented in Fig S1A and reported in the caption. We now talk of the activation of the mTORC1 pathway by NV-5297 on lines 362 and 829.

2. Line 393 and Figure 1, the authors accessed the pharmacokinetics and drug distribution in WTs (C57BL/6). However, in later experiments, N171-82Q mice were used. How to guarantee the drug behaves the same in different mouse models?

Answer 2-2: This is a valid point. The background strain of N171-82Q mice is C57BL/6 so we do not expect major pharmacodynamics discrepancies between the WT and HD mice. We now mention the N171-82Q background strain on line 409. Further, as shown in S2A, the N171-82Q treated with NV-5297 for a week present a similar pS6/S6 level in the striatum to C57BL/6 while N171-82Q treated with vehicle present a lower ratio. We also added on line 406: �We cannot discount that the pharmacokinetics of NV-5297 in the N171-82Q mice are slightly different from the controls despite the normalization of the pS6 level.

3. Line 406, the label of the figure is wrong. Figure 2 does not contain the data about pS6 levels in the striatum and heart, neither does Supplementary figure 2, which only shows a quantification of pS6/S6.

Answer 3-2: We now state clearly on line 408: �…NV-5297 treatment normalized the ratio of pS6 level in the striatum and heart to those of age-matched WT C57BL/6 mice (Fig S2A-C). �

4. Line 428 and Figure 3, the author measured the level of DARPP-32, but not the phosphorylated DARPP-32. Previous research shows that the function of DARPP-32 depends on phosphorylation and the phosphorylation sites. Plus, the function of DARPP-32 in striatum is controversial. Therefore, more evidence is required for the link between increased DARPP-32 and improved striatal health.

Answer 4-2: DARPP-32 is a specific marker of the medium spiny neurons (MSN) constituting 95% of the cell population within the striatum. DARPP-32 is a fundamental component of the dopamine-signaling cascade, and its expression is essential to the ability of dopamine to regulate the physiology of striatal neurons. Decreased DARPP-32 expression in HD starts very early in the disease and correlates with decreased neuron numbers, specifically in the striatum (Hodges et al. 2006 Human Mol Genet, Bibb et al. 2000 PNAS) and has been used in several studies including by our group (Lee et al. 2015 Neuron, Jiang et al. 2013 Human Mol Genet). We also measure the MSN size and overall striatal volume to strengthen this finding. We find that this difference can be mainly attributed to neuronal death within the striatum as the striatum is larger with similar MSN size.

5. Line 452 and Figure 4, at least one more group of WTs with vehicle treatment is required.

Answer 2-5: There is strong evidence for striatal atrophy in the N171-82Q model starting at 10 weeks (Cheng et al. 2011 NeuroImage, Aggarwal et al. 2012 NeuroImage). We also clearly established in historical data that the mTORC1 pathway activation prevents additional striatal degeneration in 13 weeks-old N171-82Q mice (Lee et al. 2015 Neuron, 2017). Respectfully, a similar result is found here and we believe is sufficient for a preliminary evaluation of NV-5297 efficiency in the context of HD. This study was set to test how NV-5297 impacts HD phenotypes, and not how NV-5297 impacts normal mice.

6. Line 470 and Figure 5, in the main context, it says that there are elevations of pS6/S6 and p4E-BP1/4E-BP1. However, there is no corresponding plots in the figure. S6 and 4E-BP1 are also missed in the representative immunoblots.

Answer 2-6: Line 476 was corrected to: ‘Post-necropsy assessment of heart tissues showed elevated pmTOR, pS6 and p4E-BP1 levels in NV-5297-treated mice versus controls (Fig 5D, E).’

7. Figure 6B, is the legend of Veh-Sal group purposely to be shorter than the others? It is hard to distinguish Veh-Sal and NV-Sal group in the figure (only Veh-Sal group is able to be seen).

Answer 2-7: Veh-Sal and NV-Sal lines overlap, making them indistinguishable. We widened the Veh-Sal group black line so that we can distinguish it more clearly on Figure 6B.

8. A. Figure 6E, the results of Veh-Sal without cardiac stress and NV-Sal without cardiac stress is different from Figure 5. There are changes in EF and FS between NV-5297 treatment group compared with vehicle group in Figure 5, but not here. Secondly, is there any difference between the Veh-Sal group with cardiac stress and the NV-Sal group with cardiac stress? Since there is no WT group, it is hard to tell which group is closer to “normal”. B. Additionally, why the increases in cardiac output and heart rate are not treated as readout of increased cardiac function? In the introduction, line 56, it says that phenotypes like smaller heart size and impaired cardiac function are exacerbated by chronic ß-adrenergic agonist in HD mouse models. However, figure 6E shows increased cardiac output and heart rate in Veh-Sal group with cardiac stress. Are the results here contradicting to the previous statement?

Answer 2-8: A. The NV-Sal and Veh-Sal groups have an osmotic pump implant on their back for the cardiac stress experiment. The presence of this pump makes the echocardiography more challenging, which justifies the difference to the baseline groups along with the smaller sample size for this experiment. For this study, we found no significant interaction of stress and cardiac stress and therefore we focused on the changes within the vehicle-treated groups and within the NV-5297-treated groups.

B. Cardiac output [CO] is calculated as the heart rate multiplied by the stroke volume [SV] (the blood volume ejected from the left ventricle per beat, line 235-239). Both Veh-Iso animals and NV-Iso animals have an increase in CO compared to baseline but through different mechanisms. The increase in CO in Veh-Iso animals is due to the increase in heart rate observed to compensate for the decreased SV observed. This decrease in stroke volume is likely due to the increase in fibrosis we observed as fibrotic tissue is stiffer and reduces muscle contractility. On the contrary, the NV-Iso animals maintain a normal heart rate but increase their cardiac output through a more efficient contraction that led to an increase in the stroke volume. This increase in stroke volume is achieved through the increase in the fractional shortening [FS] which measures the contraction as the muscle during the contraction (ventricle diameter at the end of the diastole – ventricle diameter at the end of the systole). In turn, this increase in FS lead to an increase in ejection fraction [EF] (percentage of blood ejected by the left ventricle by heartbeat). These changes can be observed visually with the echocardiogram in Fig 6D. Taken together, the CO increase in NV-Iso animals is done in a sustainable manner (physiologic hypertrophy), with the cardiac muscle growing to accommodate the increase in workload and sustain the CO while the Veh-Iso animals cope with the cardiac stress with adverse impacts (pathologic hypertrophy) such as the fibrosis, severe weight loss and increase in the mortality rate. We emphasize these effects on line 527, 530, 647 and 650.

9. Discussion Line 592, not enough evidence for the conclusion about increased MSN neurite density.

Answer 2-9: We do not conclude there is an increase in MSN neurite density, we merely suggest it as a mechanism to explain the results we have. We added in line 604: “MSN neurite density should be assessed in a follow-up study. “

---

## [Decision Letter · Decision Letter 1]

1 Aug 2022

PONE-D-22-11888R1Huntington’s disease phenotypes are improved via mTORC1 modulation by small molecule therapyPLOS ONE

Dear Dr. Davidson,

Thank you for submitting your manuscript to PLOS ONE. After careful consideration, we feel that it has merit but does not fully meet PLOS ONE’s publication criteria as it currently stands. Therefore, we invite you to submit a revised version of the manuscript that addresses the points raised during the review process.

We look forward to receiving your revised manuscript.

Kind regards,

Yuqing Li, Ph.D.

Academic Editor

PLOS ONE

Journal Requirements:

Reviewers' comments:

Reviewer's Responses to Questions

**Comments to the Author**

1. If the authors have adequately addressed your comments raised in a previous round of review and you feel that this manuscript is now acceptable for publication, you may indicate that here to bypass the “Comments to the Author” section, enter your conflict of interest statement in the “Confidential to Editor” section, and submit your "Accept" recommendation.

Reviewer #1: All comments have been addressed

Reviewer #2: (No Response)

2. Is the manuscript technically sound, and do the data support the conclusions?

Reviewer #1: Yes

Reviewer #2: Yes

3. Has the statistical analysis been performed appropriately and rigorously? 

Reviewer #1: Yes

Reviewer #2: Yes

4. Have the authors made all data underlying the findings in their manuscript fully available?

Reviewer #1: Yes

Reviewer #2: Yes

5. Is the manuscript presented in an intelligible fashion and written in standard English?

Reviewer #1: Yes

Reviewer #2: Yes

6. Review Comments to the Author

Reviewer #1: Thank you to the authors - they have adequately addressed all comments

Manuscript is suitable for publication

Reviewer #2: The authors addressed all my questions in the response to reviewers. The only concern is that the answers for questions 4 and 8 were not updated in the main context. Specifically:

1. Please add answers to question 4 about DARPP-32 to the beginning of Line 429.

2. Please justify in the main context that osmotic pump implantation in the cardiac stress experiment makes results in Figure 6 different from Figure 5, and no significant interaction of stress and cardiac stress were found so changes within the vehicle-treated groups and within the NV-5297-treated groups were focused on.

3. Please explain results in Figure 6 as described in answer 8B.

7. PLOS authors have the option to publish the peer review history of their article (what does this mean?). If published, this will include your full peer review and any attached files.

Reviewer #1: No

Reviewer #2: No

---

## [Author Response · Author response to Decision Letter 1]

6 Aug 2022

Response to Reviewers

PONE-D-22-11888

Huntington’s disease phenotypes are improved via mTORC1 modulation by small molecule therapy

Reviewer #1: Thank you to the authors - they have adequately addressed all comments

Manuscript is suitable for publication

Reviewer #2: The authors addressed all my questions in the response to reviewers. The only concern is that the answers for questions 4 and 8 were not updated in the main context. Specifically:

1. Please add answers to question 4 about DARPP-32 to the beginning of Line 429. (Question 4. Line 428 and Figure 3, the author measured the level of DARPP-32, but not the phosphorylated DARPP-32. Previous research shows that the function of DARPP-32 depends on phosphorylation and the phosphorylation sites. Plus, the function of DARPP-32 in striatum is controversial. Therefore, more evidence is required for the link between increased DARPP-32 and improved striatal health.)

Answer 1: We added at line 431: “DARPP-32 is a specific marker of the medium spiny neurons (MSN) constituting 95% of the cell population within the striatum. DARPP-32 is a fundamental component of the dopamine-signaling cascade, and its expression is essential to the ability of dopamine to regulate the physiology of striatal neurons. Decreased DARPP-32 expression in HD starts very early in the disease and correlates with decreased neuron numbers, specifically in the striatum (Hodges et al. 2006 Human Mol Genet, Bibb et al. 2000 PNAS) and has been used in several studies including by our group (Lee et al. 2015 Neuron, Jiang et al. 2013 Human Mol Genet).” We also added on line 463: “[…], suggesting that this difference can be mainly attributed to neuronal death within the striatum.”

2. Please justify in the main context that osmotic pump implantation in the cardiac stress experiment makes results in Figure 6 different from Figure 5, and no significant interaction of stress and cardiac stress were found so changes within the vehicle-treated groups and within the NV-5297-treated groups were focused on. (Question 8. A. Figure 6E, the results of Veh-Sal without cardiac stress and NV-Sal without cardiac stress is different from Figure 5. There are changes in EF and FS between NV-5297 treatment group compared with vehicle group in Figure 5, but not here. Secondly, is there any difference between the Veh-Sal group with cardiac stress and the NV-Sal group with cardiac stress? Since there is no WT group, it is hard to tell which group is closer to “normal”.)

Answer 2. Line 501: “The presence of this pump makes the echocardiography more challenging, justifying the difference to the baseline groups measurements along with the smaller sample size for this experiment. For this study, we found no significant interaction of stress and cardiac stress and therefore we focused on the changes within the vehicle-treated groups and within the NV-5297-treated groups.”

3. Please explain results in Figure 6 as described in answer 8B. (Question 8B. Additionally, why the increases in cardiac output and heart rate are not treated as readout of increased cardiac function? In the introduction, line 56, it says that phenotypes like smaller heart size and impaired cardiac function are exacerbated by chronic ß-adrenergic agonist in HD mouse models. However, figure 6E shows increased cardiac output and heart rate in Veh-Sal group with cardiac stress. Are the results here contradicting to the previous statement?)

Answer 3. We added on line 542: “CO is calculated as the heart rate multiplied by the SV (the blood volume ejected from the left ventricle per beat). Both Veh-Iso animals and NV-Iso animals have an increase in CO compared to baseline but through different mechanisms. The increase in CO in Veh-Iso animals is due to the increase in heart rate observed to compensate for the decreased SV observed. The decrease in SV is likely due to the increase in fibrosis we observed as fibrotic tissue is stiffer and reduces muscle contractility. On the contrary, the NV-Iso animals maintain a normal heart rate but increase their CO through a more efficient contraction that led to an increase in the SV. The increase in SV is achieved through increased FS which measures ventricle diameter at the end of the diastole – ventricle diameter at the end of the systole. In turn, the increase in FS causes increased EF (percentage of blood ejected by the left ventricle by heartbeat). These changes can be observed visually with the echocardiogram in Fig 6D. Taken together, the CO increase in NV-Iso animals is done in a sustainable manner (physiologic hypertrophy), with the cardiac muscle growing to accommodate the increased workload to sustain the CO, while the Veh-Iso animals do not appropriately cope with cardiac stress (pathologic hypertrophy) and show signs of fibrosis, severe weight loss and increased mortality rate.”

---

## [Decision Letter · Decision Letter 2]

12 Aug 2022

Huntington’s disease phenotypes are improved via mTORC1 modulation by small molecule therapy

PONE-D-22-11888R2

Dear Dr. Davidson,

We’re pleased to inform you that your manuscript has been judged scientifically suitable for publication and will be formally accepted for publication once it meets all outstanding technical requirements.

Kind regards,

Yuqing Li, Ph.D.

Academic Editor

PLOS ONE

Additional Editor Comments (optional):

Reviewers' comments:

Reviewer's Responses to Questions

**Comments to the Author**

1. If the authors have adequately addressed your comments raised in a previous round of review and you feel that this manuscript is now acceptable for publication, you may indicate that here to bypass the “Comments to the Author” section, enter your conflict of interest statement in the “Confidential to Editor” section, and submit your "Accept" recommendation.

Reviewer #2: All comments have been addressed

2. Is the manuscript technically sound, and do the data support the conclusions?

Reviewer #2: Yes

3. Has the statistical analysis been performed appropriately and rigorously? 

Reviewer #2: Yes

4. Have the authors made all data underlying the findings in their manuscript fully available?

Reviewer #2: Yes

5. Is the manuscript presented in an intelligible fashion and written in standard English?

Reviewer #2: Yes

6. Review Comments to the Author

Reviewer #2: (No Response)

7. PLOS authors have the option to publish the peer review history of their article (what does this mean?). If published, this will include your full peer review and any attached files.

Reviewer #2: No

---

## [Editor Report · Acceptance letter]

18 Aug 2022

PONE-D-22-11888R2 

Huntington’s disease phenotypes are improved via mTORC1 modulation by small molecule therapy 

Dear Dr. Davidson:

I'm pleased to inform you that your manuscript has been deemed suitable for publication in PLOS ONE. Congratulations! Your manuscript is now with our production department. 

Kind regards, 

on behalf of

Dr. Yuqing Li 

Academic Editor

PLOS ONE